

# Tracing North Atlantic volcanism and seaway connectivity across the Paleocene–Eocene Thermal Maximum (PETM)

Morgan T. Jones[1]*, Ella W. Stokke[1], Alan D. Rooney[2], Joost Frieling[3], Philip A.E. Pogge von Strandmann[4,5], David J. Wilson[5], Henrik H. Svensen[1], Sverre Planke[1,6], Thierry Adatte[7], Nicolas Thibault[8], Madeleine L. Vickers[1], Tamsin A. Mather[3], Christian Tegner[9], Valentin Zuchuat[10], Bo P. Schultz[11]

[1] Centre for Earth Evolution and Dynamics (CEED), University of Oslo, PO Box 1028 Blindern, 0315, Oslo, Norway
[2] Department of Earth & Planetary Sciences, Yale University, PO Box 208109, New Haven CT 06520-8109, USA
[3] Department of Earth Sciences, University of Oxford, South Parks Road, Oxford, OX1 3AN, U.K
[4] Mainz Isotope and Geochemistry Centre (MIGHTY), Institute of Geosciences, Johannes Gutenberg University, 55122 Mainz, Germany
[5] London Geochemistry and Isotope Centre (LOGIC), Institute of Earth and Planetary Sciences , University College London and Birkbeck, University of London, Gower Street, London WC1E 6BT, UK
[6] Volcanic Basin Petroleum Research (VBPR AS), Høienhald, Blindernveien 5, N-0361 Oslo, Norway
[7] Institute of Earth Sciences, University of Lausanne, 1015 Lausanne, Switzerland
[8] Department of Geosciences and Natural Resource Management, University of Copenhagen, 1350 Copenhagen K, Denmark
[9] Department of Geoscience, Aarhus University, Høegh-Guldbergs Gade 2, 8000 Aarhus C, Denmark
[10] Palaeontology and Geological Institute, Aachen University, Bergbaugebäude 1140, Wüllnerstraße 2, Aachen, Germany
[11] Museum Salling – Fur Museum, 7884 Fur, Denmark
* Corresponding author: m.t.jones@geo.uio.no

## Abstract

There is a temporal correlation between the peak activity of the North Atlantic Igneous Province (NAIP) and the Paleocene–Eocene Thermal Maximum (PETM), suggesting that the NAIP may have initiated and/or prolonged this extreme warming event. However, corroborating a causal relationship is hampered by a scarcity of expanded sedimentary records that contain both climatic and volcanic proxies. One locality hosting such a record is Fur Island in Denmark, where an expanded pre- to post-PETM succession containing hundreds of NAIP ash layers is exceptionally well preserved. We compiled a range of environmental proxies, including mercury (Hg) anomalies, paleotemperature proxies, and lithium (Li) and osmium (Os) isotopes, to trace NAIP activity, hydrological changes, weathering, and seawater connectivity across this interval. Volcanic proxies suggest that NAIP activity was elevated before the PETM and appears to have peaked during the body of the $\delta^{13}C$ excursion, but decreased considerably during the PETM recovery. This suggests that the acme in NAIP activity, dominated by flood basalt volcanism and thermogenic degassing from contact metamorphism, was likely confined to just ~200 kyr (ca. 56.0–55.8 Ma). The hundreds of thick basaltic ashes in the post-PETM strata likely represent a change from effusive to explosive activity, rather than an increase in NAIP activity. Detrital $\delta^7Li$ values and clay abundances suggest that volcanic ash production increased basaltic reactive surface area, likely enhancing silicate weathering and atmospheric carbon sequestration in the early Eocene. Signals in lipid biomarkers and Os isotopes, traditionally used to trace



paleotemperature and weathering changes, are used here to track seaway connectivity. These

proxies indicate that the North Sea was rapidly cut off from the North Atlantic in under 12 kyr

during the PETM recovery due to NAIP thermal uplift. Our findings reinforce the hypothesis that

the emplacement of the NAIP had a profound and complex impact on Paleocene–Eocene

climate, both directly through volcanic and thermogenic degassing, and indirectly by driving

regional uplift and changing seaway connectivity.

## 1. Introduction

The Paleocene–Eocene Thermal Maximum (PETM) (Kennett and Stott, 1991) was a period of

extreme global warming during the already greenhouse conditions of the early Cenozoic

(Cramwinckel et al., 2018; Zachos et al., 2008). This hyperthermal event began at ~56.0–55.9

Ma (Westerhold et al., 2017; Zeebe and Lourens, 2019) and lasted for ~150–200 kyr (Murphy et

al., 2010; Röhl et al., 2007). The PETM is characterized in the sedimentary record by a large and

sustained negative carbon isotope ($\delta^{13}$C) excursion (CIE) that varies in magnitude from 2 to 7 ‰,

with the larger excursions generally found in organic and terrestrial archives (McInerney and

Wing, 2011). The carbon cycle perturbations, and particularly the CIE onset, are traditionally

attributed to the rapid release of large volumes of $^{12}$C-enriched carbon to the ocean-atmosphere

system (Dickens et al., 1995; Zachos et al., 2008) that caused an estimated global surface

warming of ~5 °C (Dunkley-Jones et al., 2013; Frieling et al., 2017; Inglis et al., 2020). The

potential triggers of the PETM remain contentious, despite intense study of many sedimentary

sections through Paleocene–Eocene strata. Carbon sources that have received significant

attention are the destabilization of surface reservoirs such as methane hydrates (Dickens et al.,

1995), possibly triggered by orbital forcing (Li et al., 2022; Lourens et al., 2005). Other well-

studied sources include the direct volcanic emissions from the North Atlantic Igneous Province

(NAIP) (Eldholm and Thomas, 1993; Storey et al., 2007a), and thermogenic degassing from

NAIP contact metamorphism (Svensen et al., 2004).

The NAIP is a prominent candidate for the initiation and/or extended duration of the PETM

because there is a good temporal agreement between NAIP activity and the Paleocene–Eocene

boundary, and there are numerous climate forcings operating on a range of timescales associated

with its emplacement. Volatile degassing during large igneous province (LIP) eruptions and the

thermogenic release from contact metamorphism around intrusions can both release significant

volumes of carbon and sulfur to the atmosphere, directly affecting their surface cycles and



consequently the climate and environment (Jones et al., 2016). Recent studies often argue for a combination of volcanic and metamorphic NAIP sources (Gutjahr et al., 2017), or a mix of volcanic and surface reservoirs (Frieling et al., 2016) as a driver of hyperthermal conditions. Uncertainties persist because an extremely [12]C-enriched (i.e. organic-rich) source that is

sufficient to cause the CIE does not produce the magnitude of warming derived from proxy data within realistic bounds of climate sensitivity (Zeebe et al., 2009). Modelling estimates based on inverted pH proxy data arrived at far greater degassing volumes (>10,000 Gt C), which would require less [12]C-enriched sources to match the CIE (Gutjahr et al., 2017). Yet, this high-volume carbon release scenario might be at odds with the extremely enhanced organic carbon burial rates

for the PETM, a carbon sink would rapidly drive exogenic $\delta^{13}$C to positive values if not balanced by a heavily [12]C-enriched source (Kaya et al., 2022; Papadomanolaki et al., 2022). These studies suggest that the carbon source is both [12]C-enriched and voluminous, complicating interpretations of the CIE purely on grounds of the source $\delta^{13}$C signature. It is therefore imperative to constrain the volumes and fluxes of each potential carbon source to ascertain their importance for the

initiation and long duration of the PETM.

## 1.1 Direct NAIP climate impacts

### 1.1.1 Volcanic degassing

The NAIP is one of the largest known LIPs in the Phanerozoic (Ernst and Youbi, 2017), with an estimated total volume of 6–10 million km$^3$ magma emplaced at or near the Earth surface

(Eldholm and Grue, 1994; Horni et al., 2017). Assuming a magmatic $CO_2$ content of 0.5 wt% and a degassing potential of 3.5 Mt C per 1 km$^3$ of magma (Jones et al., 2016), the NAIP represents a total magmatic carbon reservoir of 21,000 to 35,000 Gt C. Much of this volume was likely degassed during effusive and explosive volcanic eruptions. While the NAIP was active from ~63–54 Ma, the main acme (~80%) of volcanism occurred from 56 to 54 Ma (Wilkinson et

al., 2017). In East Greenland, voluminous eruptions formed a ~5–6 km thick part of the flood basalt province between 56.0 and 55.5 Ma (Larsen and Tegner, 2006; Storey et al., 2007a; Storey et al., 2007b), representing a basalt accumulation rate of at least 1 cm/yr for 500,000 years. There is also evidence of significant explosive volcanism across the PETM interval from the presence of hundreds of NAIP-sourced ash layers across Northern Europe (Egger and Brückl, 2006; Jones

et al., 2019a; Larsen et al., 2003; Stokke et al., 2020b). These findings suggest that effusive and



explosive volcanic degassing from the NAIP considerably amplified global volcanic emissions of carbon and sulfur across the Paleocene–Eocene boundary.

**Figure 1**. A plate reconstruction at 56 Ma, showing the known extent of the North Atlantic Igneous Province (NAIP). Red and purple areas denote subaerial and submarine volcanism, respectively, with dark red points marking individual volcanic centres (Abdelmalak et al., 2016; Horni et al., 2017). The grey shaded areas show the known extent of NAIP sill intrusions on the continental margins (Planke et al., 2005; Rateau et al., 2013; Reynolds et al., 2017), although this is a minimum estimate as the identification of sills beneath extrusive layers is

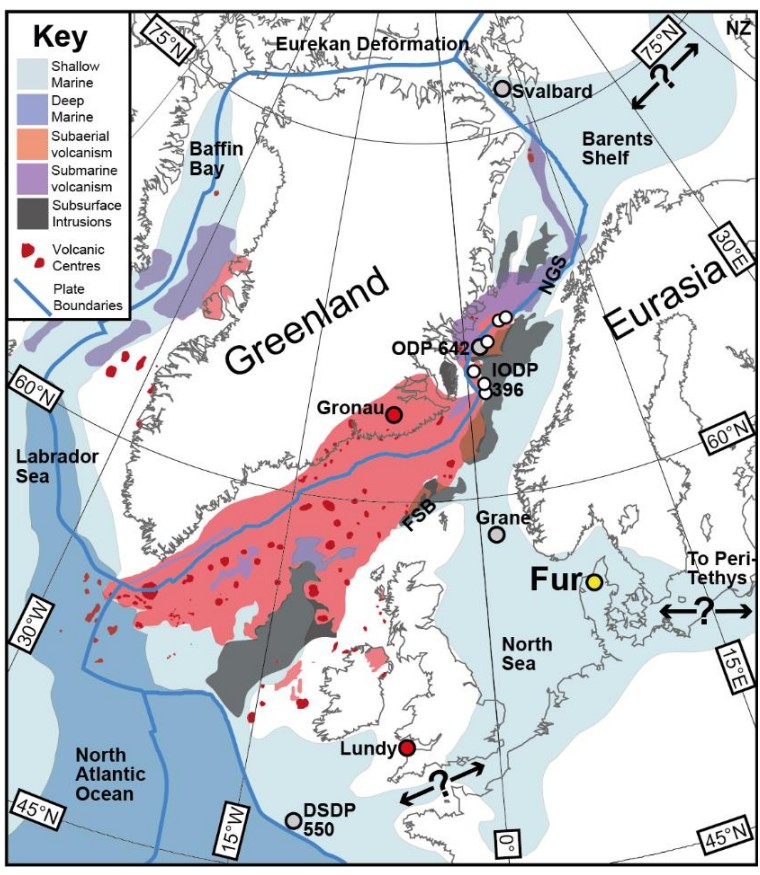

obscured by poor seismic retrievals. Palaeo-shoreline estimates are an amalgamation from several sources (Abdelmalak et al., 2016; Golonka, 2009; Hovikoski et al., 2021; Zacke et al., 2009). Shelf areas are shown in light blue, while ocean basins are shown in dark blue. NZ = Novaya Zemlya, NGS = Norwegian-Greenland Seaway, FSB = Faroe-Shetland Basin. Fur Island is labelled with a yellow marker. The volcanic localities of the Gronau West Nunatak (Heister et al., 2001) and Lundy Island (Larsen et al., 2003) have red markers. The locations of DSDP Site 550 (Goban Spur), ODP Site 642, Svalbard, and Grane cores (Jones et al., 2019a; Knox, 1984; 1985; Wieczorek et al., 2013) are shown with grey markers, while IODP Expedition 396 sites (Planke et al., 2022) are shown with white markers. The plate reconstruction was created using a modified version of GPlates (Boyden et al., 2011; Gurnis et al., 2012; Shephard et al., 2013) and plotted with Generic Mapping Tools (Wessel et al., 2013).





### 1.1.2 Thermogenic emissions

In addition to extrusive activity, massive sill complexes were emplaced in organic-rich sediments around the northeast Atlantic continental margins (Figure 1). The sill edges connect to thousands 140 of explosive hydrothermal vent complexes that were formed through volatile generation and gas-overpressure during contact metamorphism (Svensen et al., 2004). Many of these vent systems terminate at or close to the Paleocene–Eocene palaeo-surface on the Greenlandic, Faroe–Shetland, and Norwegian continental margins (Hansen, 2006; Manton et al., 2022; Planke et al., 2005; Reynolds et al., 2017). Degassing was likely a combination of magmatic gases and 145 thermogenic volatiles formed during contact metamorphism. The high methane ($CH_4$) content in thermogenic volatiles both enhances the atmospheric greenhouse effect compared to $CO_2$ and leads to more $^{12}C$-enriched emissions, making this source a prime candidate for causing the PETM CIE. Recent drilling of two hydrothermal vent complexes shows that both were formed just before or during the PETM (Frieling et al., 2016; Planke et al., 2022). Estimates for carbon 150 release from hydrothermal vents on the Norwegian continental margin range from 225 to 2250 Gt C (Svensen et al., 2004), with the degassing over the entire area affected by NAIP intrusions possibly up to 13,000 Gt C (Jones et al., 2019b). This volatile source has the potential for rapid (<100 kyr) and voluminous degassing if many sill-vent systems were active simultaneously.

### 1.2 Indirect NAIP climate impacts

In addition to volatile degassing, the emplacement of the NAIP may have affected the global climate system through other processes such as increased continental weathering and widespread regional uplift:

### 1.2.1 Enhanced carbon sinks

The PETM led to a global increase in continental weathering and erosion (Pogge von 160 Strandmann et al., 2021; Pujalte et al., 2015; Ravizza et al., 2001) and widespread enhanced marine organic carbon burial (John et al., 2008; Kaya et al., 2022; Papadomanolaki et al., 2022), which both act as negative feedbacks to increased atmospheric $CO_2$ levels. The emplacement of flood basalt lavas and widespread ash deposits would have significantly enhanced the availability of fresh reactive silicate material at the surface (Dessert et al., 2003). Volcanic ash from 165 explosive eruptions is particularly important in this process due to its high surface area-to-volume ratio (Ayris and Delmelle, 2012; Longman et al., 2021). Combined with an intensified





hydrological cycle (Carmichael et al., 2017; Walters et al., 2022), this increase in reactive volcanic substrate amplified the transport of weathered material to the oceans (Nielsen et al., 2015; Stokke et al., 2021). Enhanced fluvial fluxes of nutrients and alkalinity is likely to have

increased carbon sequestration through both carbonate formation and organic matter burial (Jones et al., 2016). The silicate weathering of NAIP basalts and ash deposits has therefore been proposed as a potential carbon sink that acted as a negative feedback to global warming and aided the termination of the PETM (Longman et al., 2021; Stokke et al., 2021).

**1.2.2 Regional uplift**

The North Sea was a pivotal epicontinental sea with intermittent connections to the Arctic, Atlantic, and Tethys oceans (Figure 1). An estimated 1–3 km of transient convective uplift occurred between Greenland and the British Isles during the latest Paleocene (Hartley et al., 2011; Shaw Champion et al., 2008; White and Lovell, 1997). The NAIP uplift has been cited as a potential source of methane hydrate release from raised marine sediments (Maclennan and Jones,

2006). This uplifted region would have had a marked effect on atmospheric and oceanic circulation, particularly seaway connectivity to the Arctic Ocean that likely had a strong influence on global climate (Roberts et al., 2009). A sporadic shallow-marine connection in the English Channel connected the North Sea with the North Atlantic Ocean (Zacke et al., 2009), whereas Paleocene NAIP uplift closed the strait in the Faroe–Shetland Basin and it remained

closed until at least 54 Ma (Hartley et al., 2011; Shaw Champion et al., 2008). Further north, thermal uplift and lava delta progradation narrowed the Norwegian–Greenland Seaway to possibly as little as 50 km of open water (Hovikoski et al., 2021), potentially isolating the North Sea from the intra-rift seaways of the northern Norwegian margin (Figure 1). While the Central Spitsbergen Basin in Svalbard was not directly connected to the Arctic due to the Eurekan

deformation (Straume et al., 2022), there was likely a broad seaway across the Barents Shelf that connected the Norwegian–Greenland Seaway with the Arctic Ocean somewhere between Svalbard and Novaya Zemlya (Prøis, 2015) (Figure 1). To the east of the North Sea, there may also have been a shallow seawater connection to the Peri-Tethys through eastern Europe (Radionova et al., 2003).

**1.3 An improved understanding of the NAIP**



Despite the close temporal link between the NAIP emplacement and the PETM, the exact relationship is complicated by multiple concurrent climate forcings, incomplete/imprecise geochronological data (Wilkinson et al., 2017), and uncertainties in the timing and sources of volatile fluxes from the NAIP (Passey and Jolley, 2009; Stoker et al., 2018). These uncertainties

are compounded by a lack of expanded sedimentary records that contain both volcanic and climatic proxies in the same sections. The limited dispersal of many geochemical/geological indicators of volcanism mean that such proxies are often regionally constrained (Jones, 2015; Jones et al., 2019a). In addition, paleoclimate proxies across the Paleocene–Eocene transition are often complicated by the significant changes in sedimentation, seawater acidification, regional

uplift/subsidence, eustatic sea level, and seaway connections. These uncertainties can only be resolved by targeted studies on expanded and continuous sedimentary sequences proximal to the NAIP that contain a multitude of volcanic and climatic proxies.

The eastern North Sea basin is an ideal setting for constraining the uplift history and magmatic activity of the NAIP across the PETM. This region has experienced near continuous tectonic

subsidence since the Late Cretaceous, resulting in high sedimentation rates and expanded sedimentary sequences. The relative proximity of this basin to the NAIP (Figure 1) resulted in the co-preservation of multiple volcanic and climatic proxies. Additionally, this location was sufficiently distant from LIP activity to avoid subtle climatic indicators being overwhelmed by the volcanic signal, and was relatively unaffected by the regional forced regressions caused by

thermal uplift. The enclosed nature of the North Sea and the lack of significant thermal or diagenetic overprints has resulted in the exceptional preservation of both inorganic and organic records (Nielsen, 1995). These factors make North Sea sediments ideal for a wide range of integrated geochemical, biological, oceanographic, and volcanological studies. The Limfjord area in Denmark (Figure 2) offers rare onshore access to North Sea Paleocene and Eocene strata

due to glaciotectonic uplift (Pedersen, 2014). These outcrops provide a unique opportunity to attain a high-quality, virtually uninterrupted record of NAIP activity and environmental change spanning the latest Paleocene and early Eocene (Heilmann-Clausen et al., 1985).

This study presents a compilation of new and existing proxy data for volcanism (ash layers, Hg anomalies, Os isotopes) and paleoclimate (lipid biomarkers, C, Li, and Os isotopes) from the

Stolleklint Beach section on Fur Island and other localities around the Limfjord area (Figure 2). These data are uniquely poised to: 1) assess changes in environmental conditions across the





PETM and in the earliest Eocene; 2) interrogate how the style and magnitude of NAIP activity
varied concurrently; and 3) evaluate how these regional signals relate to global changes through

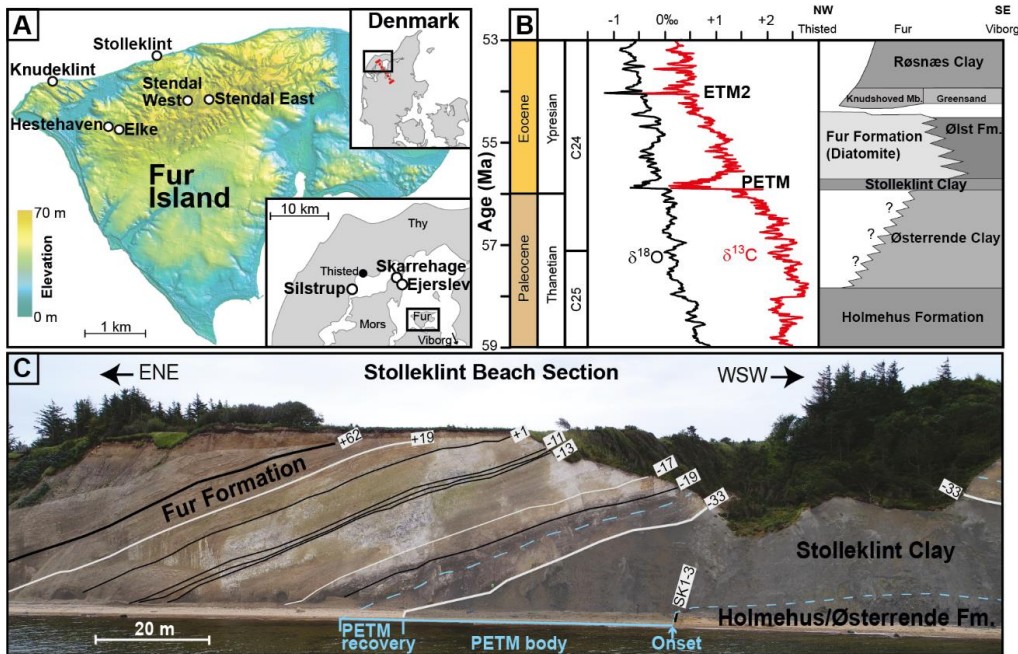

comparison to existing PETM localities worldwide.

**Figure 2**: **A)** A topographic map of Fur Island showing the locations of the Stolleklint, Knudeklint, and quarry
sections. The regional map of Limfjorden shows the Skarrehage and Ejerslev localities on Mors, and the Silstrup
locality. The Denmark map shows the line of the stratigraphic cross section in red between Thisted and Viborg. **B)** A
synthesis of the target interval, showing the Paleogene stratigraphy of northern Denmark (Heilmann-Clausen et al.,
1985; King, 2016) in relation to the GTS2020 geological timescale (Speijer et al., 2020) and global oxygen and
carbon isotope curves (Cramer et al., 2009; Littler et al., 2014). PETM = Paleocene–Eocene Thermal Maximum;
ETM2 = Eocene Thermal Maximum 2. **C)** A view of Stolleklint Beach from the north, with the outcropping strata
labelled (Stokke et al., 2021). Black lines denote key basaltic ash layers, while white lines mark the silicic ash layers
-33, -17, and +19. The onset, body, and recovery of the PETM CIE are highlighted in light blue. The boundary
between the Holmehus/Østerrende Formation and the Stolleklint Clay in the cliff face obscured by slumping is
inferred from sub-beach outcrops.

## 2. Study area

All of the samples used in this study were derived from the Paleogene succession in northern
Denmark (Figure 2). Pleistocene glaciotectonism provides valuable onshore outcrops of these
Paleogene sediments, but the associated folding and faulting often limits individual outcrops to





just part of the overall stratigraphy. However, there are over 180 NAIP ash layers (>0.2 cm
thick) found within the succession (Larsen et al., 2003), which act as marker horizons and enable
a precise correlation between outcrops and a composite reconstruction of the whole sequence of
strata. The ashes are numbered and subdivided into a negative ash series (Ash -39 to Ash -1) and
a positive ash series (Ash +1 to Ash +140) based on chemical variations and outcrop appearances

(Bøggild, 1918). The succession is divided here into nine intervals, based on changes in
lithology, the presence of ash layers, and/or bulk organic $\delta^{13}$C values.

The primary locality is Stolleklint Beach on the north coast of Fur Island, which contains a
complete outcrop of pre-PETM to early Eocene strata (Figure 2C). The base of the sequence
comprises a heavily bioturbated, hemipelagic mudstone devoid of carbonate microfossils,

referred to here as the Holmehus/Østerrende Formation (Interval 1) due to the uncertain lateral
extent of the Østerrende Clay (Figure 2B; (Heilmann-Clausen, 1995; Stokke et al., 2021). The
overlying unit is the Stolleklint Clay (Intervals 2 to 6), a ~24 m thick expanded section of dark,
laminated, thermally immature clay, almost completely devoid of benthic fauna and calcareous
fossils (Heilmann-Clausen et al., 1985). The transition from the Holmehus/Østerrende Formation

to the Stolleklint Clay is marked by a possible hiatus of unknown duration and a glauconite-rich
silty horizon (Heilmann-Clausen, 1995; Schmitz et al., 2004; Schoon et al., 2015), above which
there is no clear evidence of any breaks in sedimentation up to the top of the Fur Formation
(Heilmann-Clausen et al., 1985; 2014; Stokke et al., 2020a). Two thick ash layers (named SK1
and SK2) are found at the base of the Stolleklint Clay (Interval 2), directly below the onset of the

PETM CIE.

The sediments above Ash SK2 show the PETM onset (Interval 3), with the first appearance of
the diagnostic dinoflagellate cyst *Apectodinium augustum* (Heilmann-Clausen, 1994) and a
$\delta^{13}$C$_{org}$ excursion of -4.5 ‰ over just 11 cm (Jones et al., 2019a; Schoon et al., 2013). The body
of the PETM $\delta^{13}$C excursion comprises the bulk of the Stolleklint Clay (Intervals 4 to 6), with

$\delta^{13}$C$_{org}$ values largely stable at -31 ‰ (Jones et al., 2019a). The subdivision of the PETM body is
based on a sustained increase in TOC concentrations at -12.96 m that marks the boundary
between Intervals 4 and 5, and the re-emergence of ash layers from -5.92 m onwards that marks
Interval 6. The Stolleklint Clay is overlain by a ca. 52 m thick, fossil-rich clayey-diatomite
named the Fur Formation. The PETM $\delta^{13}$C$_{org}$ recovery (Interval 7) is ~4.5 m thick and

constrained between Ashes -33 and -21a (Heilmann-Clausen et al., 1985). The lower post-PETM



strata (Interval 8) contains occasional heterolithic ash layers of the negative ash series (Larsen et al., 2003) and the diatomite displays frequent laminations (Pedersen et al., 2004). The start of Interval 9 is marked by the appearance of Ash +1, the first of 140 mainly tholeiitic basalts of the positive ash series that corresponds to the Balder Formation offshore (King, 2016). In contrast to

much of Interval 8, the interstitial diatomites of the positive ash series (Interval 9) are mostly non-laminated (Pedersen et al., 2004).

### 3. Materials and Methods

### 3.1 Materials

The succession was logged and sampled at several outcrops and quarries on Fur Island (Jones et

al., 2019a; Stokke et al., 2020a), supplemented with Skarrehage and Ejerslev localities on Mors Island and at Silstrup on mainland Denmark (Figure 2A). These sequences were compiled to form a composite stratigraphic succession where the zero of the depth scale is set to the top of the ubiquitous and easily distinguishable felsic Ash -33 (Figure 2C). The stratigraphic thicknesses vary slightly between localities, so sections within the composite succession are tied

to specific localities. The lowermost pre-PETM to post-PETM section (-24.8 to +21.8 m) is based on the Stolleklint Beach locality (Jones et al., 2019a). The Elke and Stendal East quarries (+21.8 to +41.0 m) form the basis of the log between Ash -11 and Ash +98. The uppermost part (+41.0 to +51.0 m) is based on the Silstrup locality (Figure 2A).

The presence of abundant ash layers in the Danish strata offers insights into the extent and style

of NAIP volcanism, and act as key marker horizons between localities. High-precision radiometric dating of magmatic crystals within ash deposits provides a geochronological framework into which palaeoenvironmental records can be placed (Lowe, 2011). Key felsic ash layers in the Danish strata such as Ashes -17 and +19 are important marker horizons across Greenland and Europe (Storey et al., 2007a; Westerhold et al., 2009). Some of the basaltic ash

layers are up to 12 cm thick and found 700–1500 km from the known source volcanoes, representing some of the largest explosive basaltic eruptions recorded in the geological record (Egger and Brückl, 2006; Stokke et al., 2020b). The Stolleklint Clay and Fur Formation were systematically logged and sampled, with the thickness of each ash layer recorded and combined into a percentage of the strata (after compaction) per metre of section.


 

### 3.2 Mercury anomalies

A widely used volcanic proxy is mercury (Hg) contents in sedimentary records (Grasby et al., 2019; Percival et al., 2021). Volcanic emissions are a primary source of gaseous $Hg^0$ that is rapidly dispersed through the atmosphere (Pyle and Mather, 2003). Prolonged periods of

elevated volcanism, such as from LIPs, directly impact the global Hg cycle (Grasby et al., 2019). Mercury has a strong affinity to organic matter, which provides the main depositional pathway for Hg in aqueous environments (Outridge et al., 2007). Elevated Hg/TOC (total organic carbon) is interpreted to show enhanced Hg sequestration, either through enrichment of Hg in organic matter or via deposition through other means such as adhered to clay particles or in sulfides

(Sanei et al., 2012). Therefore, anomalously high Hg content or peaks in host-phase normalised ratios (e.g. Hg/TOC, Hg/S) can be indicative of extensive regional or global volcanic activity (Grasby et al., 2019).

High resolution sampling and analyses were conducted for Hg and TOC through the Stolleklint Clay (Jones et al., 2019a) and the Fur Formation (this study). The lowermost strata (-24.8 to -

24.3 m) was continuously sampled, while the mean sample spacing for the rest of the stratigraphy was 8 cm (including 425 new samples). All samples were oven dried at ≤50 °C and powdered either in an agate hand mortar or agate disc mill before further analysis. Mercury contents were analysed using a Zeeman R-915F (Lumex) high-frequency atomic absorption spectrometer at the University of Lausanne. Samples were heated to >700 °C and measurements

were taken on the direct thermal evaporation of Hg from solid samples. Each aliquot was measured in duplicate, while machine accuracy was confirmed by the analysis of the GSD-11 standard certified reference material (Chinese alluvium: 72.0 ±3.6 ppb). Total organic carbon (TOC) concentrations were obtained by Rock-Eval pyrolysis (Behar et al., 2001) at the University of Lausanne.

### 3.3 Lithium isotopes

The ratio of lithium isotopes $^7Li$ and $^6Li$, expressed as $\delta^7Li$, is a tracer for silicate weathering in sedimentary records. Lithium is abundant in silicate rocks, largely absent from carbonate rocks, and is not known to be fractionated by organic growth (Kisakürek et al., 2005; Pogge von Strandmann et al., 2016). As such, Li isotopes are dominantly controlled by silicate weathering

processes (Pogge von Strandmann et al., 2021). In fluvial waters, Li isotopes reflect the balance between primary rock dissolution and secondary clay formation. Isotopic fractionation through



clay formation increases dissolved $\delta^7Li$ values, whereas primary rock dissolution does not fractionate Li isotopes and therefore drives dissolved $\delta^7Li$ to low values (Misra and Froelich, 2012; Pogge von Strandmann et al., 2017b). Low-intensity fluvial weathering regimes (i.e. high

erosion relative to weathering rates) are characterized by high primary rock dissolution relative to clay formation, leading to low dissolved $\delta^7Li$ values but high dissolved Li fluxes. Moderate-intensity weathering regimes have greater clay formation and therefore elevated dissolved $\delta^7Li$ values, but somewhat lower dissolved Li fluxes. High-intensity weathering regimes induce the dissolution of secondary clays with little primary rock dissolution, which leads to low $\delta^7Li$

values and low Li fluxes (Dellinger et al., 2015; Pogge von Strandmann et al., 2020). In this study, we analysed the solid (largely clay) material from the sediment sequences, providing a counterpart to the dissolved signal. The advantage compared to marine carbonate sections is that detrital clay sections are not affected by the long (~1 Myr) ocean residence time of Li, but instead inform on local conditions in the catchment area above the depositional site on much

more rapid timescales (Dellinger et al., 2017; Pogge von Strandmann et al., 2017a; 2021). These characteristics make Li isotopes a powerful tracer for changes in the hydrological cycle through surface runoff, as well as silicate weathering intensity, which directly impacts atmospheric $CO_2$ drawdown fluxes.

Selected samples were analysed through the Stolleklint Clay (Pogge von Strandmann et al.,

2021) and the Fur Formation (this study). Powdered samples were dissolved first using HF-$HNO_3$-$HClO_4$, then steps of concentrated $HNO_3$ and 6M HCl. The samples were then purified for Li using a two column method with AG 50W-X12 resin and 0.2M HCl as an eluent (Pogge von Strandmann et al., 2017a). The purified samples were analysed at the LOGIC group at University College London using a Nu Plasma 3 multi-collector inductively-coupled plasma mass

spectrometer (MC-ICP-MS) and normalised to IRMM-016 bracketing standards. Long-term external analytical uncertainties using this method are ± 0.4‰ (2σ) (Pogge von Strandmann et al., 2021).

### 3.4 Clay mineralogy

Samples from the Fur Formation were analysed for clay mineralogy to expand on the existing

dataset from Stokke et al. (2021). The clay fraction (<2 μm) was prepared as oriented aggregate mounds using gravity settling and Millipore filter transfer method (Moore and Reynolds, 1997).



The XRD clay data were recorded with a step size of 0.01 from 2 to 65 (2 θ) at a count time of
0.3 s (2 θ) in air-dried samples, and with a step size of 0.01 from 2 to 34 (2 θ) at a count time of
0.3 s (2 θ) on treated samples. Three rounds of treatments were applied: 24 h of ethylene glycol
saturation, 1 h heating at 350 °C, and 1 h heating at 550 °C. The software NewMod II (Reynolds
III and Reynolds Jr., 2012) was used for semi-quantification of the XRD patterns of inter-
stratified clay minerals.

### 3.5 Rhenium and osmium isotopes

The osmium ($^{187}Os/^{188}Os$) isotope system is a powerful paleoceanographic tracer for silicate
weathering, specifically identifying changes between evolved and juvenile sources through time
(Peucker-Ehrenbrink and Ravizza, 2000; Peucker-Ehrenbrink and Ravizza, 2020). The ratio of
rhenium to osmium (Re/Os) is higher in crustal rocks than in the mantle, which means the *in situ*
beta decay of $^{187}Re$ to $^{187}Os$ leads to higher (radiogenic) $^{187}Os/^{188}Os$ ratios over time. The crustal
$^{187}Os/^{188}Os$ average is ~1.40 (Peucker-Ehrenbrink and Jahn, 2001), while the unradiogenic
primitive upper mantle value is ~0.13 (Meisel et al., 2001). The oceanic record reflects the
proportional mixing of these two end-members (Peucker-Ehrenbrink and Ravizza, 2000), with
the unradiogenic component sourced from the weathering and alteration of subaerial basalts,
submarine hydrothermal activity (Dickson et al., 2021), and occasional extra-terrestrial bolide
impacts (Sato et al., 2013).

Osmium is removed from the water column through the precipitation of Fe–Mn oxides and/or
adsorption onto organic material and clays (Racionero-Gómez et al., 2017; Yamashita et al.,
2007). The oceanic residence time of Os is 10–55 kyr (Levasseur et al., 1999; Sharma et al.,
1997), which is longer than the present-day mixing time of the oceans (1–2 kyr) but considerably
shorter than for other weathering tracers such as strontium (>2 Myr) (Broeker and Peng, 1982).
While the global oceans are relatively homogeneous at a given time, $^{187}Os/^{188}Os$ ratios can
deviate rapidly from the fully mixed global signal in restricted environments such as enclosed
basins (Rooney et al., 2016). Enhanced organic matter deposition can reduce Os residence times
significantly, and local input and output fluxes become more important in enclosed settings
(Martin et al., 2001; Paquay and Ravizza, 2012). Therefore, the $^{187}Os/^{188}Os$ proxy has multiple
potential uses in this study, including tracking the emplacement and weathering of NAIP lavas
and ashes, changes in continental weathering associated with regional uplift or climatic



variations, and modifications to the extent of seaway connectivity between the North Sea basin and the Arctic, Tethys, and Atlantic realms.

Individual samples weighing ca. 50 g were selected for Os isotope chemostratigraphy analysis throughout the succession. Consolidated samples were cut using a rock saw and then hand-polished with a diamond-plated polishing pad to remove potential contamination from the saw blade and weathered surfaces. After being dried at room temperature overnight, all samples were crushed to a fine (~30 μm) powder in a SPEX 8500 Shatterbox with a ceramic agate grinding container and puck to homogenize any potential Re and Os heterogeneity. The long half-life of
$^{187}$Re (ca. 42 Gyr) leads to minimal age correction for Cenozoic samples, but both Re and Os concentrations and isotopic compositions were measured in this study to provide accurate initial Os isotope compositions for chemostratigraphy.

Sample Re and Os isotopic abundance and composition were determined at the Yale Geochemistry and Geochronology Center. Depending on Re concentration, between 0.3 and 1.0
g of sample powder was digested and equilibrated in 8 ml of $Cr^{VI}O_3$-$H_2SO_4$ with a mixed $^{190}$Os-$^{185}$Re tracer (spike) solution sealed in Carius tubes at 220°C for 48 hours. This dissolution method has been shown to preferentially liberate hydrogenous Re and Os to yield a more accurate and precise depositional age (Kendall et al., 2004). The Re and Os were extracted and purified through solvent extraction (NaOH, $(CH_3)_2CO$, and $CHCl_3$), micro-distillation, anion
column chromatography, and analysed using negative thermal ionization mass spectrometry (Selby and Creaser, 2003). Isotopic measurements were performed via static Faraday collection for Re, and ion-counting using a secondary electron multiplier in peak-hopping mode for Os, on a ThermoElectron TRITON PLUS mass spectrometer (Creaser et al., 1991; Völkening et al., 1991). The Os samples were loaded onto 99.995% Pt wire (H-Cross, NJ) in 9 N HBr, covered
with a saturated solution of $Ba(OH)_2$ in 0.1 N NaOH as activator, and analysed as oxides of Os. Interference of $^{187}$ReO$_3$ on $^{187}$OsO$_3$ was corrected using the measured intensity of $^{185}$ReO$_3$. Mass fractionation was corrected with $^{192}$Os/$^{188}$Os = 3.0826, using the exponential fractionation law.

In-house Re and Os solutions were continuously analysed during this study to ensure and monitor long-term mass spectrometer reproducibility. The Yale Geochemistry and
Geochronology Center Re standard solution measured on the Faraday cups yields an average $^{185}$Re/$^{187}$Re value of 0.59748 ± 0.0014 (2σ, n = 6), which agrees with the accepted value within



error (Gramlich et al., 1973). The measured difference in $^{185}$Re/$^{187}$Re values for the Re solution

and the accepted value (0.59738) is used to correct the Re sample data. The Os isotope standard

solution used at Yale Metal Isotope Center is the Durham Romil Osmium Standard (DROsS)

(Luguet et al., 2008). Over the past three years on the Yale Triton, the runs yield a $^{187}$Os/$^{188}$Os

ratio of 0.16082 ± 0.000116 (2σ, n = 319) that is identical, within uncertainty, to the value

reported by other laboratories (e.g. Liu and Pearson, 2014; Luguet et al., 2008). Total procedural

blanks during this study were 40 ± 2.0 pg for Re and 0.06 ± 0.11 pg for Os, with an average

$^{187}$Os/$^{188}$Os value of 0.25 ± 0.05 (1σ).

Uncertainties for $^{187}$Re/$^{188}$Os and $^{187}$Os/$^{188}$Os are determined by error propagation of

uncertainties in Re and Os mass spectrometry measurements, blank abundances and isotopic

compositions, spike calibrations, and reproducibility of standard Re and Os isotopic values. The

Re-Os isotopic data, 2σ calculated uncertainties for $^{187}$Re/$^{188}$Os and $^{187}$Os/$^{188}$Os are used to

generate initial Os isotope (Os$_i$) compositions with an age of 55.9 Ma.

**3.6 Seaway connectivity proxies**

The extent of North Sea isolation may be tested using existing datasets of Os isotopes (see

above) and thaumarchaeal membrane lipid distributions. The composition of isoprenoid glycerol

dialkyl glycerol tetraether lipids (GDGTs) is thought to regulate membrane fluidity of marine

Thaumarcheota (Schouten et al., 2013). As such, GDGT composition is commonly assumed to

be largely governed by growth temperature and forms the basis of the TEX$_{86}$ paleotemperature

proxy (Schouten et al., 2002). Typically, the number of cyclo-pentane rings increases with

temperature, with the TEX$_{86}$ ratio presenting the statistically strongest relation (Kim et al., 2010;

Schouten et al., 2002). However, other metrics, such as the ring index (Zhang et al., 2016) and

fractional abundance of crenarchaeol regio-isomer (fcren') relative to total crenarchaeol (O'Brien

et al., 2017), can be employed to detect non-thermal impact on TEX$_{86}$ (Zhang et al., 2016) and

differences in temperature response between communities (e.g. Inglis et al., 2015; O'Brien et al.,

2017). Indeed, it has been recognized that certain modern ocean regions have distinct GDGT

distributions and responses to temperature, arguably associated with distinct, isolated

communities of Thaumarchaea (e.g. Trommer et al., 2009). Some studies have turned this around

to argue for distinct communities based on GDGT distributions in (semi-) restricted settings

(Steinig et al., 2020).



High fcren' has been recognized in (warm) saline waters (Steinig et al., 2020; Trommer et al., 2009), and has been proposed to identify such paleo-environmental conditions (Inglis et al., 2015). In contrast, many fresh-water bodies are marked by reduced fcren' (e.g. Blaga et al.,

2009; Powers et al., 2010). At times during the latest Paleocene-early Eocene, reduced salinity water-masses occupied the North Sea area (Bujak and Mudge, 1994; Eldrett et al., 2014; Kender et al., 2012; Zacke et al., 2009) and the Arctic Ocean (Pagani et al., 2006; Sluijs et al., 2006). Here we re-examine the published Paleocene–Eocene GDGT data, including those from the North Sea area, focusing on the differences in response to PETM warming between TEX$_{86}$ and

fcren' to assess whether GDGT distributions can be used as a supporting tool to detect basin restriction and/or reduced salinity.

## 4. Results

### 4.1 Volcanic ash layers

The NAIP ash layers are largely constrained to specific ash-rich intervals within the Stolleklint

Clay and Fur Formation. Four ashes (SK1–SK4) are found just above and below the PETM onset (Figure 3B), representing the earliest evidence of explosive volcanism in Danish strata (Heilmann-Clausen et al., 2014). There is a distinct lack of ash layers for a ~19.3 m interval between the PETM onset and Ash -39, the first of the traditional numbered ash series (Figure 3A). The second interval of ash-rich strata (-4.9 to +10.9 m) encompasses the recovery of the

δ$^{13}$C$_{org}$ excursion and includes several felsic ash layers such as Ashes -33 and -17. It is followed by a ~15 m thick ash-poor interval, with only three ash layers >1 cm thickness. Ash +1 at +26.0 m heralds the main phase of ash deposition in the Fur Formation (Figure 3A). With the exception of the felsic +13 and +19 ashes, the positive ash series comprises relatively homogenous tholeiitic basalts. Basaltic ash reaches a peak of 31% of the total sediment (after compaction)

between +42.0 and +43.0 m (Ashes +110 to +118; Figure 3A), representing a considerable portion of the total stratigraphy.









**Figure 3**: Composite data from the studied outcrops. **A)** The full succession from late Paleocene to Eocene strata, with the zero on the depth scale set to the top of Ash -33. **B)** The first metre of the condensed Stolleklint Beach section. Note the change in scale for some proxies. The glauconite-rich horizon is marked in the lower stratigraphic log, along with ash layers SK1 to SK4 (Stokke et al., 2021). The age model is based on four marker horizons: the PETM onset at 55.93 Ma (Westerhold et al., 2017), a 101 kyr PETM body duration (van der Meulen et al., 2020), the corrected Ar-Ar age of 55.48 ±0.12 Ma for Ash -17 (Storey et al., 2007a), and an estimated age of ~55.28 Ma for Ash +19 based on an estimated ~200 kyr interval between Ashes -17 and +19 (Röhl et al., 2007; Westerhold et al., 2009). The top of the Fur Formation is older than 54.6 Ma (King, 2016). The 'Ash thicknesses' column shows the percentage of sediment (after compaction) that is volcanic ash for each metre of strata. Black bars denote basaltic ashes, while grey bars denote felsic ashes. The $\delta^{13}C_{org}$ data are from previous studies (Jones et al., 2019a; Schoon et al., 2013), as are the sea surface temperature (SST) estimates using the $TEX_{86}$ proxy (Schoon et al., 2015; Stokke et al., 2020a; Vickers et al., 2020). Bottom water temperature estimates are from clumped isotopes of glendonite calcite (Vickers et al., 2020). The mercury (Hg) and total organic carbon (TOC) concentrations are a combination of this study and Jones et al. (2019a). Lithium ($\delta^7Li$) isotopes are from this study and Pogge von Strandmann et al. (2021). The osmium (Os) isotopes show initial $^{187}Os/^{188}Os$ values at 55.9 Ma ($Os_i$). The succession is divided into nine distinct intervals based on lithological changes and variations in $\delta^{13}C_{org}$ values: 1) the late Paleocene Holmehus/Østerrende Formation; 2) the pre-PETM; 3) the PETM onset; 4) the lower part of the PETM body; 5) the middle part of the PETM body; 6) the ash-rich upper part of the PETM body; 7) the PETM recovery; 8) the lower part of the Fur Formation; and 9) the upper part of the Fur Formation (see also Table 1).

## 4.2 Sedimentary mercury

Mercury concentrations vary from 2 to 303 ppb through the succession, but within each interval Hg contents are relatively consistent (Table 1). Mercury peaks are few and limited in amplitude, with only four samples exceeding 100 ppb. Mercury content was also normalised to TOC and plotted as Hg/TOC (Figure 3). Samples where TOC < 0.2 wt.% were excluded from Hg/TOC ratios as the propagated error creates unacceptably high uncertainties (Grasby et al., 2019). The intervals where this occurs are in the glauconite-rich layer at the top of Interval 1 (Figure 3B), and in part of the diatomite-rich Fur Formation (Intervals 7–9; Figure 3A). The late Paleocene and pre-PETM strata (Intervals 1 and 2) have relatively low mean Hg contents of 32.5 and 30.0 ppb, respectively (Table 1). When Hg is normalised to TOC, the late Paleocene strata is relatively uniform (Interval 1), while the pre-PETM strata (Interval 2) shows more scatter and sporadic Hg/TOC anomalies (Figure 3B). Mercury content increases slightly across the PETM onset (Interval 3) to an average of 43.8 ppb (Table 1), but a concurrent increase in TOC content leads to lowered Hg/TOC values compared to the interval below.

Within the PETM body, Hg becomes increasingly enriched up stratigraphy, with the largest mean concentrations observed in Interval 6 (67.7 ppb; Table 1). This concentration is comparable to the average shale value (62.4 ppb) from a compilation of global datasets (Grasby et al., 2019). In the lower and middle parts of the PETM (Intervals 4 and 5), the Hg content





covaries with TOC enrichments, resulting in an average Hg/TOC ratio of 33.8 ppb/wt.% (σ = 10.0) for Interval 4 and 26.9 ppb/wt.% (σ = 3.6) for Interval 5. The low standard deviations

highlight that Hg/TOC values are remarkably uniform through Intervals 4 and 5 (Table 1). The relative homogeneity ends in the final 5.6 m of the PETM body (Interval 6), where Hg enrichments outpace increased TOC contents, coincident with the re-emergence of ash layers in the stratigraphy (Figure 3A). The PETM recovery (Interval 7) heralds a decrease in Hg and TOC concentrations into the diatomitic Fur Formation. Mercury contents rise again slightly into

Interval 8, becoming more pronounced in the section between Ashes -17 and +1 with a mean Hg content of 47 ppb (Table 1). Variable TOC enrichments in Interval 8 lead to a large scatter in Hg/TOC ratios (Figure 3A). Mercury and TOC contents gradually decline into Interval 9, reaching a nadir (mean 21.4 ppb Hg) in the strata between Ashes +19 and +118, with many samples <0.2 wt% TOC. Mercury and TOC contents slightly increase again towards the top of

the section exposed at Silstrup, but Hg/TOC ratios are comparable to much of the rest of the Fur Formation (Figure 3A).

| Interval | Interval | L. Depth (m) | U. Depth (m) | Thickness (m) | Hg (ppb) | | | | Hg/TOC (ppb/wt%) | | | | $Hg_a$ (ng/cm²/yr) |
|---|---|---|---|---|---|---|---|---|---|---|---|---|---|
| | | | | | Min | Mean | Max | σ | Min | Mean | Max | σ | |
| 1 | late Paleocene | -24.82 | -24.59 | 0.23 | 9 | **32.5** | 54 | 10.0 | 48 | **78.1** | 164 | 23.7 | ? |
| 2 | pre-PETM | -24.58 | -24.38 | 0.20 | 19 | **30.0** | 40 | 7.2 | 64 | **86.8** | 130 | 23.9 | ? |
| 3 | PETM onset | -24.37 | -24.26 | 0.11 | 22 | **40.4** | 81 | 14.2 | 30 | **58.1** | 158 | 45.4 | **0.20** |
| 4 | Lower PETM body | -24.25 | -12.97 | 11.28 | 25 | **37.1** | 110 | 14.6 | 24 | **33.8** | 78 | 10.3 | **1.41** |
| 5 | Middle PETM body | -12.96 | -5.93 | 7.03 | 54 | **60.4** | 72 | 6.1 | 21 | **26.9** | 33 | 3.6 | **2.30** |
| 6 | Upper PETM body | -5.92 | 0.00 | 5.92 | 38 | **67.7** | 227 | 35.4 | 14 | **51.7** | 324 | 61.6 | **2.58** |
| 7 | PETM recovery | 0.01 | 4.50 | 4.49 | 17 | **37.6** | 78 | 16.0 | 19 | **84.6** | 164 | 36.3 | **0.13** |
| 8 | Fur Fm lower | 4.51 | 25.99 | 21.48 | 9 | **44.8** | 303 | 26.7 | 14 | **102.5** | 608 | 68.8 | **0.32** |
| 9 | Fur Fm upper | 26.00 | 51.06 | 25.06 | 2 | **28.1** | 67 | 12.7 | 13 | **61.4** | 152 | 25.3 | ? |

**Table 1**: Compilation of the average Hg contents (ppb), Hg/TOC ratios (ppb/wt.%), and Hg accumulation rates ($Hg_a$) for the nine distinct intervals of the stratigraphy.


### 4.3 Lithium isotopes

There are significant variations in detrital Li isotopes ($\delta^7$Li) through the studied section. Within the late Paleocene and pre-PETM strata (Intervals 1 and 2), $\delta^7$Li values are typically between -1.1 and +0.2‰. The two samples encompassing the negative $\delta^{13}$C onset (at -24.37 and -24.26 m)

show a -3.9‰ $\delta^7$Li excursion (Figure 3B), accompanied by a TEX$_{86}$-based ~10 °C sea surface temperature (SST) warming across the same interval (Schoon et al., 2015; Stokke et al., 2020a). The Li isotopes gradually return to less negative values through the course of the PETM body, returning to -0.7‰ by the end of Interval 6 (Figure 3A). Lithium isotope values then oscillate




from -2.4 to +0.2 and back to -2.2‰ through the PETM recovery (Interval 7). The post-PETM

strata (Interval 8) first shows a gradual positive $\delta^7Li$ excursion, peaking at 0.5‰ at ~1.5 m above

Ash -17 (Figure 3A). There is then a progressive negative $\delta^7Li$ excursion to -4.0‰ just above

Ash +9. The remaining samples from the upper Fur Formation between Ashes +31 and +98

(Interval 9) have $\delta^7Li$ values between -1.8 and -1.4‰.

## 4.4 Rhenium and osmium isotopes

| Depth | Interval | Interval | TOC (wt.%)* | Re (ppb) | ± | Os (ppt) | ± | $^{192}Os$ (ppt) | ± | $^{187}Re/^{188}Os$ | ± | $^{187}Os/^{188}Os$ | ± | $Os_i$ (55.9 Ma) | ± |
|---|---|---|---|---|---|---|---|---|---|---|---|---|---|---|---|
| -24.81 | 1 | Late Paleocene | 0.42 | 4.59 | 0.03 | 142.8 | 0.5 | 56.3 | 0.3 | 162.4 | 1.4 | 0.498 | 0.003 | **0.347** | 0.008 |
| -24.76 | 1 | Late Paleocene | 0.42 | 3.70 | 0.02 | 146.0 | 0.7 | 57.3 | 0.4 | 128.4 | 1.0 | 0.523 | 0.006 | **0.403** | 0.009 |
| -24.71 | 1 | Late Paleocene | 0.42 | 3.30 | 0.01 | 209.5 | 0.6 | 84.2 | 0.3 | 77.9 | 0.4 | 0.341 | 0.002 | **0.268** | 0.007 |
| -24.63 | 1 | Late Paleocene | 0.33 | 4.68 | 0.04 | 172.5 | 0.5 | 68.3 | 0.3 | 136.2 | 1.2 | 0.450 | 0.003 | **0.324** | 0.007 |
| -24.60 | 1 | Late Paleocene | 0.21 | 4.30 | 0.02 | 156.7 | 0.6 | 62.6 | 0.4 | 136.9 | 1.1 | 0.386 | 0.003 | **0.258** | 0.007 |
| -24.59 | 1 | Late Paleocene | 0.19 | 2.13 | 0.01 | 117.4 | 0.3 | 46.9 | 0.2 | 90.5 | 0.6 | 0.385 | 0.002 | **0.301** | 0.008 |
| -24.57 | 2 | Pre-PETM | 0.27 | 3.32 | 0.01 | 218.6 | 0.6 | 87.7 | 0.3 | 75.3 | 0.4 | 0.352 | 0.002 | **0.282** | 0.008 |
| -24.56 | 2 | Pre-PETM | 0.33 | 4.06 | 0.01 | 184.4 | 0.6 | 73.8 | 0.3 | 109.4 | 0.6 | 0.370 | 0.002 | **0.269** | 0.007 |
| -24.53 | 2 | Pre-PETM | 0.33 | 3.19 | 0.01 | 211.4 | 0.4 | 84.9 | 0.2 | 74.7 | 0.2 | 0.336 | 0.001 | **0.267** | 0.007 |
| -24.45 | 2 | Pre-PETM | 0.44 | 2.73 | 0.01 | 185.1 | 0.6 | 74.2 | 0.4 | 73.1 | 0.5 | 0.354 | 0.003 | **0.286** | 0.008 |
| -24.44 | 2 | Pre-PETM | 0.33 | 3.15 | 0.01 | 209.5 | 0.6 | 84.2 | 0.3 | 74.6 | 0.4 | 0.341 | 0.002 | **0.271** | 0.005 |
| -24.37 | 3 | PETM onset | 0.51 | 196.39 | 0.47 | 500.6 | 1.9 | 156.4 | 0.3 | 2498.2 | 8.0 | 2.593 | 0.007 | **0.265** | 0.005 |
| -24.35 | 3 | PETM onset | 0.38 | 31.70 | 0.06 | 310.4 | 0.9 | 117.8 | 0.3 | 535.3 | 1.8 | 0.801 | 0.003 | **0.302** | 0.007 |
| -24.26 | 3 | PETM onset | 1.15 | 23.96 | 0.05 | 383.2 | 1.1 | 147.8 | 0.4 | 322.6 | 1.1 | 0.670 | 0.003 | **0.370** | 0.005 |
| -24.05 | 4 | PETM body | 0.92 | 17.88 | 0.03 | 343.2 | 0.8 | 132.9 | 0.3 | 267.7 | 0.7 | 0.635 | 0.002 | **0.386** | 0.012 |
| -23.55 | 4 | PETM body | 1.14 | 23.55 | 0.05 | 362.5 | 0.9 | 139.7 | 0.3 | 335.2 | 1.0 | 0.672 | 0.002 | **0.359** | 0.019 |
| -22.30 | 4 | PETM body | 0.96 | 25.80 | 0.05 | 353.9 | 0.9 | 136.0 | 0.3 | 377.3 | 1.1 | 0.695 | 0.002 | **0.343** | 0.013 |
| -19.84 | 4 | PETM body | 1.26 | 36.71 | 0.07 | 434.9 | 1.0 | 165.4 | 0.3 | 441.6 | 1.1 | 0.783 | 0.002 | **0.371** | 0.017 |
| -17.31 | 4 | PETM body | 1.16 | 22.24 | 0.04 | 312.0 | 0.7 | 119.4 | 0.2 | 370.5 | 1.0 | 0.732 | 0.002 | **0.386** | 0.012 |
| -14.17 | 4 | PETM body | 1.25 | 27.29 | 0.05 | 310.7 | 0.7 | 117.7 | 0.2 | 461.3 | 1.2 | 0.818 | 0.002 | **0.388** | 0.013 |
| -12.37 | 5 | PETM body | 1.78 | 34.49 | 0.07 | 383.5 | 0.9 | 145.8 | 0.3 | 470.6 | 1.2 | 0.786 | 0.002 | **0.347** | 0.012 |
| -10.48 | 5 | PETM body | 2.24 | 40.02 | 0.08 | 450.5 | 0.9 | 170.7 | 0.3 | 466.4 | 1.2 | 0.816 | 0.002 | **0.381** | 0.013 |
| -8.56 | 5 | PETM body | 2.61 | 44.65 | 0.09 | 513.6 | 1.1 | 194.8 | 0.3 | 456.0 | 1.1 | 0.807 | 0.002 | **0.382** | 0.012 |
| -6.58 | 5 | PETM body | 1.78 | 37.04 | 0.09 | 417.6 | 1.1 | 158.9 | 0.3 | 463.6 | 1.5 | 0.779 | 0.002 | **0.347** | 0.012 |
| -2.12 | 6 | PETM body | 3.14 | 74.10 | 0.14 | 591.7 | 1.3 | 219.7 | 0.3 | 670.9 | 1.6 | 0.984 | 0.002 | **0.359** | 0.014 |
| -0.78 | 6 | PETM body | 3.90 | 76.44 | 0.15 | 644.5 | 3.5 | 242.1 | 1.4 | 628.1 | 3.7 | 0.886 | 0.010 | **0.301** | 0.016 |
| 0.31 | 7 | PETM recovery | 0.70 | 6.93 | 0.03 | 560.5 | 2.3 | 210.5 | 0.9 | 65.5 | 0.4 | 0.888 | 0.006 | **0.826** | 0.014 |
| 1.71 | 7 | PETM recovery | 0.91 | 6.61 | 0.01 | 96.6 | 0.4 | 35.2 | 0.2 | 374.0 | 1.8 | 1.159 | 0.006 | **0.811** | 0.017 |
| 7.65 | 8 | Early Eocene | 3.22 | 189.10 | 0.37 | 795.4 | 3.9 | 265.4 | 0.9 | 1417.6 | 5.5 | 1.947 | 0.011 | **0.626** | 0.015 |
| 15.00 | 8 | Early Eocene | 1.09 | 12.71 | 0.02 | 1009.4 | 3.2 | 366.7 | 0.9 | 69 | 0.2 | 1.174 | 0.004 | **1.110** | 0.013 |
| 27.05 | 9 | Early Eocene | 0.32 | 0.91 | 0.01 | 250.1 | 0.6 | 98.4 | 0.2 | 18.4 | 0.1 | 0.506 | 0.002 | **0.490** | 0.020 |
| 34.00 | 9 | Early Eocene | 0.41 | 1.03 | 0.01 | 54.2 | 0.2 | 21 | 0.1 | 97.8 | 0.6 | 0.636 | 0.004 | **0.540** | 0.014 |
| 41.00 | 9 | Early Eocene | 0.54 | 3.48 | 0.01 | 215.8 | 0.7 | 79.6 | 0.2 | 86.9 | 0.3 | 1.04 | 0.004 | **0.960** | 0.022 |


**Table 2**: Rhenium and osmium geochemistry for the studied strata. *Total organic carbon (TOC) content (Jones et al., 2019a) shown for reference.

The Re and Os abundances and isotopic compositions vary considerably in the studied samples.

Elemental Re and Os abundances range from 0.9 to 196.4 ppb, and 54 to 1009 ppt, respectively

(Table 2). $^{187}Re/^{188}Os$ ratios vary between 18 and 2498, while $^{187}Os/^{188}Os$ ratios vary from 0.336

to 2.593. Rhenium and Os enrichments appear to covary with TOC content, with the PETM

interval showing elevated element abundances compared to pre- and post-PETM strata (Table 2).

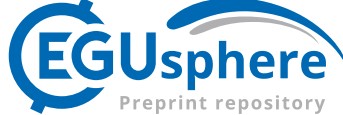

Using an assigned age of 55.9 Ma, calculated initial $^{187}Os/^{188}Os$ values ($Os_i$) range from 0.258 to 1.110, with significant and systematic changes through the stratigraphy. The late Paleocene

samples (Interval 1) are consistently unradiogenic, with $Os_i$ values between 0.26 and 0.40 (Table 2; Figure 3B). The $Os_i$ values are lower in the ash-bearing pre-PETM strata (Interval 2), consistently between 0.27 and 0.29. The PETM onset (Interval 3) is marked by an increase in $Os_i$ values from 0.27 to 0.39. Throughout the body of the $\delta^{13}C$ excursion (Intervals 4–6), the $Os_i$ values remain unradiogenic (0.30 – 0.39) and remarkably stable for ~24 m of stratigraphy (Table

2). However, the PETM recovery samples (Interval 7) exhibit an extreme change in $Os_i$ values from 0.3 to 0.83 across a ~1 m interval containing the felsic Ash -33 (Figure 3A). This more radiogenic signature persists throughout the Fur Formation, albeit with considerable variations in $Os_i$ (0.49–1.11).

### 4.5 Geochronology and accumulation rates

Absolute age estimates allow for the quantification of sediment deposition rates for environmental proxies such as TOC and Hg. A robust geochronological framework is achievable for the Stolleklint Clay due to the clear onset, body, and recovery of the PETM $\delta^{13}C$ excursion, and for parts of the Fur Formation due to the prevalence of ash layers that act as key marker horizons (Westerhold et al., 2009). In contrast, the chronology of the pre-PETM strata is poorly

constrained due to intense bioturbation and a potential unconformity below the glauconite-rich horizon.

Sedimentation rates during the PETM can be estimated by two methods. The first method uses the estimated durations of the PETM onset and body from astronomically-calibrated localities and modelling. If the 11 cm of preserved strata is the entire PETM onset (Figure 3B), and

assuming a maximum duration of 5 kyr for this interval (Kirtland Turner, 2018), the minimum average sedimentation rates at the start of the hyperthermal were ~2.2 cm/kyr (Table 1). For the PETM body, van der Meulen et al. (2020) predict a CIE duration of $101 \pm 9$ kyr. Including propagated uncertainties, this duration gives an average sedimentation rate of $23.8 \pm 4.5$ cm/kyr, up to an order of magnitude greater than the sedimentation rate during the onset interval (Table

1). The alternative method is to use marker horizons of known ages. The PETM onset is constrained by astronomical tuning to 55.93 Ma, based on the best-fit orbital solution (Westerhold et al., 2017). A bentonite horizon within the PETM CIE in Svalbard has a $^{206}Pb/^{238}U$



age of 55.785 ± 0.034 Ma (Charles et al., 2011), estimated to be equivalent to ~2 m above Ash -33 in the Danish strata (Stokke et al., 2020a). Using these two ages gives a sedimentation rate of

17.9 ± 2.8 cm/kyr, including uncertainties of the thickness of the Stolleklint Clay and the position of the Svalbard marker horizon in the Fur strata. The slight disparity between these two sedimentation rate estimates may arise because the latter value includes part of the PETM recovery, where deposition rates were reduced at Fur. Overall, the two methods are in good agreement and we base our age model on the former estimate.

Estimating Fur Formation deposition rates is aided by the presence of numerous ash layers, which have been correlated with tuff layers within the East Greenland flood basalts at the Gronau Nunatak (Heister et al., 2001; Storey et al., 2007a), offshore along the Northwest Europe continental margin of Goban Spur (DSDP Site 550), and the Norwegian continental margin (ODP Site 642; Figure 1) (Knox, 1984; Knox, 1985). In particular, two prominent felsic ash

layers (Ashes -17 and +19) are used to constrain the geochronology of the early Eocene. Ash -17 was Ar-Ar dated to 55.48 ± 0.12 Ma, once corrected using the 28.201 Ma Fish Canyon Tuff calibration (Kuiper et al., 2008; Storey et al., 2007a). Astronomical tuning using records from Site 550 and the Walvis Ridge (ODP Sites 1262/1263) predicts a ~200 kyr interval between the two ashes (Röhl et al., 2007; Westerhold et al., 2009), which gives an estimated age of ~55.28

Ma for Ash +19 (Figure 3). The measured thickness between Ash -17 and Ash +19 based on the Stolleklint and Elke outcrops is 18.67 m, or 18.02 m excluding ash intervals, which gives an estimated deposition rate of 9.0 cm/kyr for this section of the Fur Formation (Table 1).

## 5. Discussion

### 5.1 Constraining NAIP activity

The main proxies for the eruption style and intensity of NAIP activity are ash deposition, Os isotope chemostratigraphy, clay mineralogy, and Hg anomalies, and each of these proxies has its strengths and weaknesses. Ash layers are conclusive evidence of explosive eruptions. However, the eastern North Sea is 700–1500 km from the known NAIP source volcanoes (Figure 1), which means that only the largest explosive eruptions are preserved in Danish strata. Several ashes

show evidence of magma–water interactions in their formation (Stokke et al., 2020b), suggesting that hydro-magmatic processes led to the explosive nature of eruptions. Therefore, while the distribution of ashes is a clear indicator of extreme explosive volcanism, it may reflect changes





in eruption style and may not be indicative of overall volcanic activity. Osmium isotopes can be used as a passive tracer for the weathering and erosion of LIP basalts and ashes affecting ocean

chemistry, providing the signal is discernible from other factors such as changing continental weathering and other sources of unradiogenic Os such as extra-terrestrial inputs and mid-ocean ridge spreading (Dickson et al., 2021). Another indicator of the weathering and erosion of LIP basalts and ash is the presence of the minerals smectite and zeolite (Nielsen et al., 2015; Stefánsson and Gíslason, 2001). While clay mineralogy is affected by changes in climate and

hydrology, the presence of both smectite and zeolite minerals in variable abundance throughout the Fur stratigraphy has been linked mainly to weathering of volcanic products of NAIP origin (Heilmann-Clausen et al., 1985; Stokke et al., 2021), rather than to reworking of smectite-rich sediments (Li et al., 2020) or post-deposition flocculation (Deconinck and Chamley, 1995).

Sedimentary mercury is generally accepted as a viable proxy for large scale volcanism in the

geological record, with Hg and Hg/TOC anomalies coeval with periods of (subaerial) LIP emplacement (Grasby et al., 2019). However, other factors can impact Hg and TOC contents non-uniformly (Grasby et al., 2019; Percival et al., 2015), so it is important to rigorously assess individual localities taking these factors into account. Directly comparing (normalised) Hg records between localities is complicated by differences in depositional environments, such as

organic matter source and content, lithology, sediment accumulation rates, and redox state (Grasby et al., 2019). It is also currently challenging to differentiate between thermogenic and volcanic sources for Hg in the sedimentary record, since both are large potential Hg sources. The regional distribution of Hg emissions would be heavily affected by whether Hg degassing is subaerial or submarine (Jones et al., 2019a; Percival et al., 2018). Thermogenic degassing from

the NAIP was most likely dominated by shallow marine venting (Svensen et al., 2004), which may suggest that Hg anomalies proximal to the NAIP are more likely to be of contact metamorphic origin. However, given that widespread sill intrusions and continental flood basalts are both chronologically constrained to somewhere in this time interval, and that the ratio of submarine versus subaerial Hg emissions for each source is poorly constrained, the Hg record

can only give an overall qualitative indicator of NAIP activity across this interval.

Existing data across the PETM suggests larger Hg anomalies closer to the NAIP (Jones et al., 2019a; Keller et al., 2018; Kender et al., 2021; Liu et al., 2019; Tremblin et al., 2022), indicating that subaqueous emissions from volcanic and/or thermogenic sources may well have limited Hg





deposition to more proximal settings than from atmospheric distribution. In fact, even within the
North Sea basin there appears to be a substantial gradient in the magnitude and frequency of Hg
anomalies, decreasing in intensity from northwest to southeast (Jones et al., 2019a; Kender et al.,
2021). Therefore, the Danish strata does not experience significant Hg anomalies compared to
more proximal localities. However, what the Fur succession does have is an expanded
sedimentary sequence and a well-constrained geochronology, which allows us to estimate Hg
mass accumulation rates ($Hg_a$) using the equation (1):

$$Hg_a = (Hg_c)(SR)\rho$$

$$Hg_a(ng/cm^2/yr) = Hg_c\left(\frac{ng}{g}\right) \times SR\left(\frac{cm}{yr}\right) \times \rho\left(\frac{g}{cm^3}\right)$$

where $Hg_c$ is the mean measured Hg concentration, $SR$ is the estimated sedimentation rate, and $\rho$
is the density of the host sediments. The density of the Stolleklint Clay is measured as 1.4 g/cm³
while the Fur Formation diatomite has a measured density of 0.8 g/cm³ (Pedersen et al., 2004).
The calculated $Hg_a$ values for each interval are shown in Table 1.

### 5.1.1. Late Paleocene (Holmehus/Østerrende Formation), Interval 1

There are no ash deposits in the Holmehus or Østerrende clays in Denmark (Heilmann-Clausen
et al., 2014), but occasional ash layers are present in the coeval Lista Formation in the North Sea
(Haaland et al., 2000; Knox and Morton, 1988). Both Hg and TOC are relatively uniform up
until the base of the glauconite-rich horizon (Figure 3B). Mean Hg/TOC values are 78.1
ppb/wt% (Table 1), comparable to the average of published shale datasets (Grasby et al., 2019).
Global ocean $^{187}Os/^{188}Os$ values were low during the late Paleocene (Os$_i$ ≈ 0.4), considerably
more unradiogenic than present day values of 1.06 (Dickson et al., 2021). These values imply
that basalt weathering was already a major component of global Os fluxes, both from ongoing
NAIP activity and the earlier tropical emplacement of the Deccan Traps at 66.5–65 Ma (Schoene
et al., 2019). Late Paleocene sediments in the North Sea have abundant smectite and zeolite
mineral components (Nielsen et al., 2015; Stokke et al., 2021), potentially indicating extensive
weathering and denudation of basaltic material into the epicontinental sea. The Os$_i$ values in the
Holmehus/Østerrende Formation are variable between 0.40 and 0.27, becoming more
unradiogenic towards the glauconite-rich horizon (Table 2; Figure 3). This finding suggests that
low level NAIP activity was occurring in the late Paleocene, and potentially increasing into the



latest Paleocene, consistent with the estimated magmatic activity of the NAIP at this time
(Wilkinson et al., 2017).

**5.1.2. Pre-PETM (Stolleklint Clay), Interval 2**

The latest Paleocene section at Fur is a condensed interval of just 20 cm, 12 cm of which are the
ash layers SK1 and SK2 (Figure 3B). These ashes mark the first explosive eruptions that were
large enough to reach Denmark (Heilmann-Clausen et al., 2014). The interstitial sediments
between the thick ash layers also appear to have a large ash component (Stokke et al., 2021).

Mercury concentrations are not elevated with respect to the underlying strata, but Hg/TOC
values are consistently high with a mean of 86.8 ppb/wt% (Table 1; Figure 4). Osmium isotopes
are extremely low in this interval, with a mean $Os_i$ of 0.275 (Table 2; Figure 3B). A pre-PETM
shift to unradiogenic $^{187}Os/^{188}Os$ values has been noted elsewhere, particularly at Svalbard
(Wieczorek et al., 2013) and Millville, New Jersey (Liu et al., 2019). This apparent global pulse

of unradiogenic Os to the marine realm was most likely of NAIP origin (Dickson et al., 2021), as
the negative $^{187}Os/^{188}Os$ excursions in Svalbard and Fur are accompanied by Hg anomalies in the
interstitial sediments that predate the PETM CIE (Jones et al., 2019a). In addition, the clay
mineral smectite comprises 38% of the mineralogy in Interval 2 at Fur (Heilmann-Clausen et al.,
1985; Stokke et al., 2021), suggesting a significant input of weathered basaltic material.

**5.1.3. PETM onset (Stolleklint Clay), Interval 3**

The PETM onset at Fur is bounded by ash SK2 and the much thinner SK3 and SK4 layers that
are the last ash occurrences for >19 m of strata (Figure 3), suggesting that explosive volcanism is
ongoing but waning. Trace metals typically associated with volcanic ash such as Ni, Cu, and V
are not enriched in the sediments in this interval (Stokke et al., 2021). Aside from the sample

directly above ash SK2, there are no Hg anomalies and Hg/TOC values are reduced compared to
pre-PETM strata (Table 1; Figure 3B). Calculated Hg deposition rates ($Hg_a$, using equation 1)
are 0.12 ng/cm²/yr, assuming the onset is complete and 5 kyr in duration (Figure 5). This rate is
less than the Hg deposition rates of 0.7 to 1.6 ng/cm²/yr calculated for the unpolluted Holocene
(<1000 CE) Baltic Sea (Frieling et al., in review), but greater than the estimated 0.05 to 0.1

ng/cm²/yr of atmospheric Hg deposition in Holocene Swedish peat bogs (Bindler, 2003). The
lack of Hg anomalies during the PETM onset may suggest a period of volcanic quiescence, or
perhaps a shift to subaqueous Hg emissions (either volcanic or thermogenic) became more





dominant, thereby affecting more NAIP-proximal localities than Denmark (Jones et al., 2019a; Kender et al., 2021).

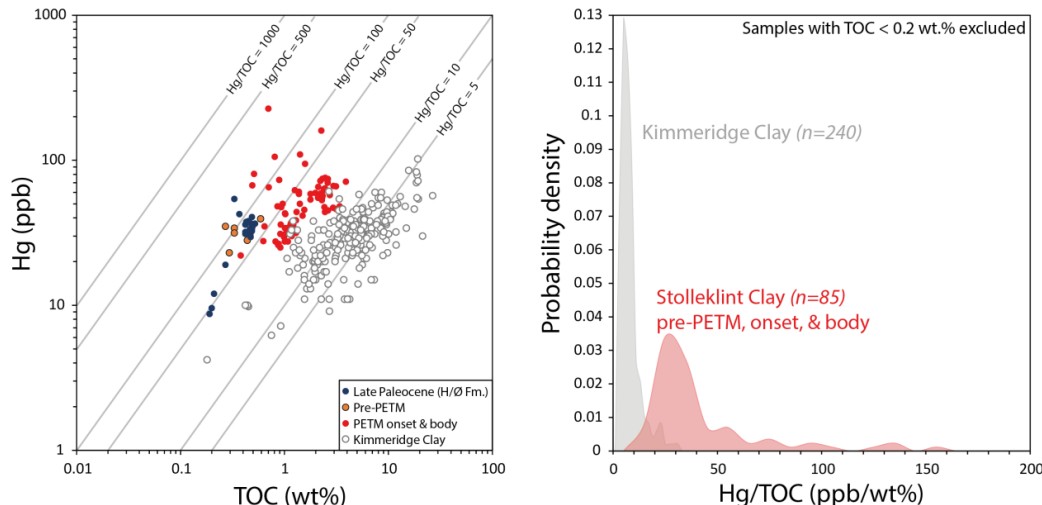

**Figure 4.** Left: A comparison of mercury (Hg) vs total organic carbon (TOC) data for the Holmehus/Østerrende formation (late Paleocene) and the Stolleklint Clay (pre-PETM, onset and body) compared with the Upper Jurassic Kimmeridge Clay (southern England) that is an organic-rich shale control site that was not associated with LIP volcanism (Percival et al., 2015). Right: A probability density diagram for Hg/TOC ratios in the Stolleklint Clay (red) and the Kimmeridge Clay (grey).


The Os isotope record shows a pronounced (>0.1) radiogenic shift through the PETM onset (Figure 3B), which has also been noted at other localities worldwide (Dickson et al., 2015; Liu et al., 2019; Ravizza et al., 2001; Wieczorek et al., 2013). This change has been interpreted to record enhanced continental weathering in response to hyperthermal conditions and a more

vigorous hydrological cycle (Dickson et al., 2015; Pujalte et al., 2015; Ravizza et al., 2001). This hypothesis is corroborated by large changes in clay assemblages, Li isotopes, and surface temperature proxies across the CIE onset at Fur (Figure 3B) (Pogge von Strandmann et al., 2021; Schoon et al., 2015; Stokke et al., 2021; Stokke et al., 2020a). The shift to radiogenic $^{187}$Os/$^{188}$Os values suggests that an increase in submarine volcanism is unlikely during the onset, but the

radiogenic shift could reflect large scale thermogenic release of Os to the ocean-atmosphere system during NAIP contact metamorphism (Dubin, 2015). The rapidity of the change in Os isotopes is also notable, as the onset duration (Kirtland Turner, 2018) is less than the modern





oceanic residence time of Os (10–55 kyr (Levasseur et al., 1999; Sharma et al., 1997). Changes on such a timescale suggest that not only did Os inputs change across the PETM onset, but also

that marine Os export may have increased rapidly due to enhanced sedimentation rates and organic matter burial.

### 5.1.4. PETM body (Stolleklint Clay), Intervals 4–6

Evidence for explosive volcanic activity during the ~24 m PETM body is limited to the lowermost 12 cm (ashes SK3 and SK4) and uppermost 5 m (Ashes -39 to -33) of strata. The

sediments in between (Intervals 4 and 5) have no macroscopic ashes. However, a large zeolite component (up to 36.1% of bulk mineralogy) in this interval suggests continued weathering of volcanic material in sediment source areas (Stokke et al., 2021), and Os isotopes show a mantle-dominated signature of ~0.4 (Figure 3). The Hg and TOC signal in this interval is noteworthy as their covariance results in remarkably stable Hg/TOC ratios (Figure 3), indicating that the Hg

cycle was coupled to TOC deposition and was in steady state. The Hg/TOC values are lower than both the pre- and post-PETM sequences (Table 1), which at first glance suggests low volcanic activity. However, when compared to the late Jurassic Kimmeridge Clay, which was deposited under similar conditions (i.e., anoxic, high productivity) during a period with no LIP activity (Percival et al., 2015), the Hg/TOC ratios in the Stolleklint Clay are considerably

elevated (Figure 4). It is also conceivable that the rapid sedimentation rates and high TOC contents led to suppressed Hg/TOC anomalies during the PETM.

Using a sedimentation rate of 23.8 cm/kyr, based on an estimated PETM body duration of 101 ± 9 kyr (van der Meulen et al., 2020), and a mean Hg content of 52.8 ppb through Intervals 4–6 gives an estimated Hg mass accumulation rate ($Hg_a$) of 1.76 ng/cm$^2$/yr. This rate is an order of

magnitude higher than the estimated $Hg_a$ during the PETM onset, and considerably higher than those calculated for the Holocene (Bindler, 2003; Frieling et al., in review). The general trend through the PETM body is of increasing Hg contents up-section, which outpace concomitant TOC increases in Interval 6, leading to the highest mean Hg content and discrete Hg/TOC anomalies coincident with the re-emergence of ash layers in the stratigraphy (Figure 3).

Assuming constant sedimentation rates, we calculate $Hg_a$ to be 1.24 ng/cm$^2$/yr for Interval 4 (lower PETM body), 2.01 ng/cm$^2$/yr for Interval 5 (middle PETM body), and 2.26 ng/cm$^2$/yr for Interval 6 (upper PETM body; Table 1; Figure 5). If an average $Hg_a$ of 1.76 ng/cm$^2$/yr during the





PETM body is indicative of the whole North Sea basin at this time (~500,000 km$^2$), it would

result in annual deposition of 8.8 tonnes of Hg into this epicontinental sea, and a total of ~0.9 Mt

Hg deposition across the duration of the PETM body. This is likely a conservative estimate,

given that more proximal sites have higher Hg contents, and that >20 m deposition during the

PETM body is not unusual in this basin (Jin et al., 2022). The calculated Hg accumulation flux

for the North Sea is 1.3–12% of current (albeit poorly-constrained) estimates of modern global

volcanic Hg emissions (Grasby et al., 2019; and references therein), despite representing just

0.1% of the Earth's surface area.

**Figure 5.** Volcanic proxies through the studied section, normalised to their estimated depositional age based on the age model shown in Figure 3. **A)** Carbon isotopes ($\delta^{13}C_{org}$) for reference (Jones et al., 2019a), showing the PETM CIE from 55.93–55.83 Ma. **B)** Ash thicknesses as a combined percentage of ash (after compaction) per metre of sediment. Black bars denote basaltic ashes and the grey bars denote felsic ashes. **C)** Initial $^{187}Os/^{188}Os$ isotopes ($Os_i$). **D)** Estimated Hg accumulation rates ($Hg_a$), subdivided into specific intervals. The volcanic proxy records below the glauconite-rich layer are not

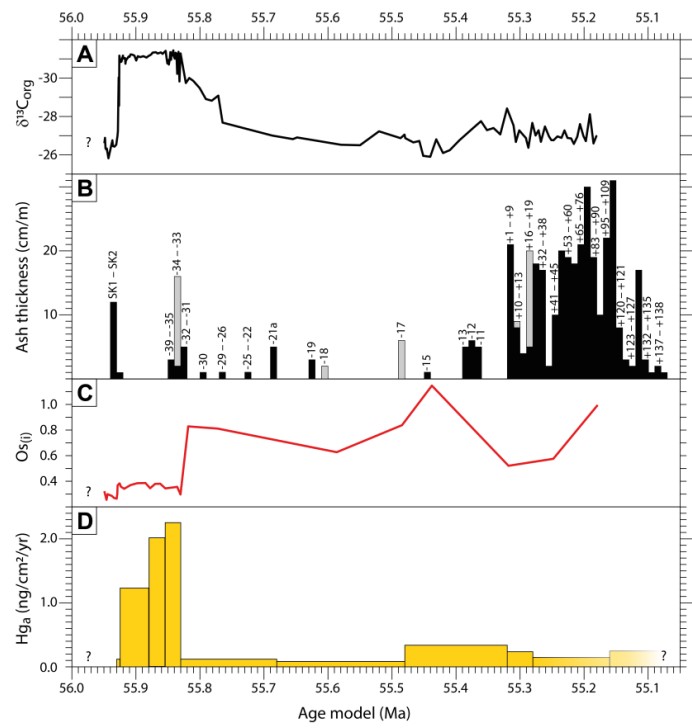

shown due to uncertainties in the timing of deposition, while $Hg_a$ values in the youngest part of the strata are uncertain due to poorly constrained sedimentation rates above Ash +19.

High Hg deposition rates during the CIE body are substantiated by sediments in Svalbard, where

Hg content and Hg/TOC ratios are consistently elevated during the PETM compared to overlying

and preceding strata (Jones et al., 2019a). Svalbard is unique as a PETM locality in that there

were high sedimentation rates before, during, and after the CIE, with no breaks in sedimentation

nor significant changes in lithology (Charles et al., 2011; Dypvik et al., 2011), so the potential




for sedimentological disruption to the Hg signal is minimised. The combined records from Fur Island and Svalbard suggest that Hg emissions were substantially elevated during the PETM, resulting in higher Hg accumulation rates in these rapidly deposited sediments (Figure 3). The

increase in $Hg_a$ values and Hg/TOC anomalies towards the end of the CIE body at Fur (Interval 6), combined with the reappearance of ash layers in the stratigraphy, may suggest an increase in NAIP activity. However, it could potentially be an artefact of assuming constant sedimentation rates through the PETM body, whereas both Li isotope variations (Pogge von Strandmann et al., 2021) and clay mineral assemblages (Stokke et al., 2021) suggest that surface runoff and erosion

rates were greater earlier in the PETM. This discrepancy may affect the individual $Hg_a$ interval estimations, but does not change the main finding that average Hg accumulation rates during the PETM CIE were significantly elevated (Figure 5).

### 5.1.5. PETM recovery (Fur Formation), Interval 7

The PETM recovery is bounded by the thick felsic Ash -33 and the basaltic Ash -21a (Figure 3),

based on dinoflagellate cyst assemblages (Heilmann-Clausen, 1994) and $\delta^{13}C$ values (Jones et al., 2019a). Using estimated ages of ~55.83 Ma for the end of the PETM body (van der Meulen et al., 2020) and ~55.48 Ma for the radiometric age of Ash -17 (Storey et al., 2007a) gives an estimated duration of 160 kyr between Ashes -33 and -21a (Figure 3). However, the relatively well-constrained stratigraphic sections (PETM body and Ash -17 to Ash +19) are not necessarily

well correlated to each other, leading to potentially significant errors in estimating the duration of Interval 7. Although there are twelve ash layers in this interval, most are <1 cm (Bøggild, 1918) and are extremely heterogeneous in composition, suggesting sources from the NW European shelf (Larsen et al., 2003). The bulk rock mineralogy shows the zeolite component declining from 27% to 0% from -0.28 to +5.35 m depth (Stokke et al., 2021), although there is a

concurrent increase in the clay fraction from 4 to 17% that is dominated by illite-smectite (Figure 6). These mineralogical shifts suggest a fundamental change in the volcanogenic sediment fluxes into the eastern North Sea basin across the PETM recovery. The $^{187}Os/^{188}Os$ data for Intervals 7–9 is considerably more radiogenic than Intervals 1–6, suggesting a fundamental shift in Os supply into the North Sea that may not solely reflect changes in weathering of continental and

basaltic substrates (see *Section 5.2.1*).



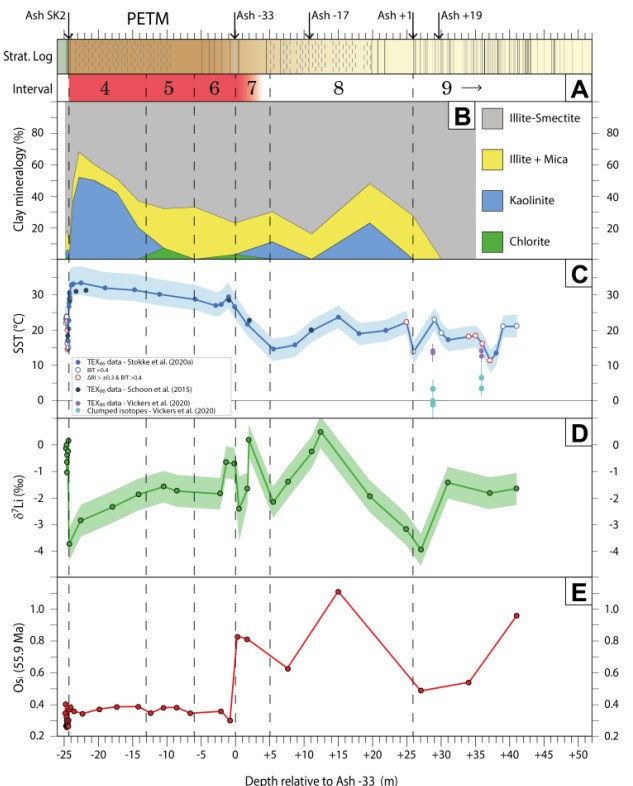

**Figure 6.** A compilation of weathering and climate proxies through the studied section. **A)** The stratigraphic log is adapted from Stokke et al. (2020a), showing laminated intervals (dashed lines) and ash layers (solid lines). Key ash layers are labelled on the log, along with the position of the PETM CIE and Intervals proposed in this study. **B)** Variations in the relative abundance of clay minerals for Intervals 1–8 (Stokke et al., 2021) and Interval 9 (this study). **C)** Sea surface temperature data based on the $TEX_{86}$ proxy (Schoon et al., 2015; Stokke et al., 2020a) and benthic temperatures based on clumped isotopes (Vickers et al., 2020). **D)** Detrital $\delta^7Li$ values from Intervals 1–7 (Pogge von Strandmann et al., 2021) and Intervals 7–9 (this study). **E)** Initial $^{187}Os/^{188}Os$ ($Os_i$) values through the studied section.

The Hg signal during the PETM recovery is likely affected by the change from clay-rich to diatomite-rich sediments and the associated change in sedimentation rates and more oxidising

conditions. Mercury contents show a general decrease from ~60 to ~20 ppb (Figure 3), but the parallel decrease in TOC is significantly greater, leading to higher Hg/TOC ratios (Table 1). The upper part of Interval 7 has TOC contents <0.2 wt%, which is taken as the cut-off for propagated errors in Hg/TOC ratios being too high. However, even at TOC contents above 0.2 wt% there is a (subtle) remaining trend that is not accounted for by a linear Hg-TOC relation, leading to

higher Hg/TOC at comparatively lower TOC in many datasets (Grasby et al., 2019). Calculating $Hg_a$ is complicated by the lack of a good chronology for this interval, coupled with a gradual change in sediment density into the diatomite-rich strata. A 160 kyr duration between Ashes -33 and -21a gives a sedimentation rate of 2.8 cm/kyr. Taking the mean Hg content of 37.6 ppb (Table 1), and assuming a density between that of the Stolleklint Clay and the Fur Formation (ρ

≈ 1.1 g/cm³), gives a mass accumulation rate ($Hg_a$) of 0.12 ng/cm²/yr. Even with the substantial





errors ingrained in this estimation it is clear that Hg deposition decreased significantly during the PETM recovery in the eastern North Sea (Figure 5). A decline in volcanogenic sediment input is corroborated by the extreme radiogenic shift in $Os_i$ isotopic values from 0.3 to 0.83 (Table 2; Figure 3A), although the amplitude of this change may be indicative of local changes in the basin
configuration rather than just changes in volcanic activity (see *Section 5.2.1*).

**5.1.6. Early Eocene (Fur Formation), Intervals 8–9**

The post-PETM Fur Formation is separated into two intervals based on the abundance and chemistry of ash layers. In Interval 8, the Ashes -21a to -1 are relatively sparse and chemically heterolithic, likely being sourced from a mixture of shelf sources and failed or propagating parts
of the central rift system (Larsen et al., 2003). In contrast, Interval 9 is dominated by vast volumes of chemically homogenous tholeiitic basaltic ashes numbered +1 to +140 (Larsen et al., 2003). Physical and chemical evidence of glass shards suggests that the central rift system started to develop surface water bodies, leading to hydro-magmatic activity and a shift from effusive to explosive activity (Stokke et al., 2020b). The clay fraction in the Fur Formation is dominated by
smectite, reaching 100% in Interval 9 (Figure 6), which suggests that the weathering and erosion of volcanic rocks into the North Sea basin continued to be an important sediment source into the early Eocene.

Mercury contents in the Fur Formation display a gradual decrease from a mean value of 47 ppb in the second part of Interval 8 (Ashes -17 to +1) to 21 ppb in the middle part of Interval 9
(Ashes +19 to +118; Table 1), before increasing slightly at the top of the section in conjunction with higher TOC contents (Figure 3). Organic carbon content is predominantly <1 wt% in the diatomite, leading to significant scatter in Hg/TOC values. The apparent enrichment in Hg/TOC compared to the Stolleklint data likely reflects the bias of inflating Hg/TOC ratios at lower TOC contents (Grasby et al., 2019). Mercury accumulation rates ($Hg_a$) reach a nadir of 0.09 ng/cm$^2$/yr
in the first part of Interval 8 (Table 1; Figure 5), although it may be partly due to the aforementioned uncertainties in sedimentation rate between Ashes -33 and -17. The better constrained sedimentation rates in the 200 kyr interval between Ashes -17 and +19 give calculated $Hg_a$ values of 0.34 ng/cm$^2$/yr for the second part of Interval 8 and 0.23 ng/cm$^2$/yr for the first part of Interval 9 (Table 1). These accumulation rates are well below the calculated $Hg_a$
values during the PETM (Figure 5) due to reduced sedimentation rates and lower density of the





diatomite. The sedimentation rates for the upper part of Interval 9 are not known, but if the calculated value of 9.0 cm/kyr between Ashes -17 and +19 is assumed to continue, then $Hg_a$ values are 0.15 ng/cm$^2$/yr between Ashes +19 and +118, and 0.25 ng/cm$^2$/yr between Ashes +118 to +140 (Table 1). These $Hg_a$ estimates indicate that the highest Hg deposition rates occur

in the second half of Interval 8, and not concomitant with the start of the voluminous explosive volcanism in Interval 9 (Figure 5).

### 5.2 Tracing seaway connectivity

The Paleocene–Eocene transition is marked by several regional and eustatic changes in sea levels. A global sea-level rise on the order of a few metres during the PETM is attributed to the

thermal expansion of seawater (Sluijs et al., 2008). While this effect would be important for shallow slope/margin environments, it is insufficient to affect seaway connectivity. In contrast, the transient thermal uplift in the centre of the NAIP closed the Atlantic connection via the Faroe-Shetland basin until ~54 Ma (Hartley et al., 2011; Shaw Champion et al., 2008; White and Lovell, 1997). Evidence from the $\delta^{18}$O record of shark-tooth apatite indicates a North Sea

surface-water freshening in the early Eocene (Zacke et al., 2009), suggesting that the Atlantic connection via the English Channel was also temporarily restricted. This North Sea freshening is corroborated by an influx of low-salinity tolerant dinocyst taxa into the North Sea prior to the PETM CIE (Kender et al., 2012). There is also evidence of a transient closure of the shallow marine connection to the Peri-Tethys (Radionova et al., 2003) and a restriction of the

Norwegian-Greenland Seaway (Hovikoski et al., 2021) around the late Paleocene to early Eocene. Although the exact timings of these events are not well constrained, these observations indicate the potential for restriction of the proto-Northeast Atlantic basins and the North Sea. The multiple datasets from the Danish Paleogene strata in this study and prior work can be compared with other global high-resolution records to explore changes to the oceanic connections between

the North Sea and the Atlantic, Tethys, and Arctic Oceans.

### 5.2.1. Osmium isotopes

The existing $^{187}$Os/$^{188}$Os data across the PETM is largely homogeneous across the North Atlantic, Indian, and Tethys Oceans (Figure 7) (Dickson et al., 2015; Liu et al., 2019; Ravizza et al., 2001; Schmitz et al., 2004). Overall, the global marine $^{187}$Os/$^{188}$Os pool reflects a highly

unradiogenic signature (Os$_i$ ≈ 0.4) before, during, and after the PETM (Figure 7). There is some





variability between localities, such as a pronounced drop in $Os_i$ in pre-PETM strata from the North Atlantic that suggests elevated unradiogenic fluxes prior to the onset of the CIE (Liu et al., 2019; Schmitz et al., 2004). This signal appears to be absent from other sites, but that may be due to data gaps and/or condensed sections (Figure 7). The late Paleocene, pre-PETM, and PETM

body intervals in Denmark have $Os_i$ values between 0.258 and 0.403 (Table 2; Figure 5), with the more unradiogenic values found in pre-PETM strata. These values are in good agreement with global ocean datasets, which suggests that there was an open seaway connection between the North Sea and the North Atlantic Ocean before and during the PETM CIE (Figure 1). In contrast to the global ocean signal, the $Os_i$ records from Svalbard (Wieczorek et al., 2013) and

Lomonosov Ridge (Dickson et al., 2015) show much more radiogenic signatures (Figure 7). This heterogeneity suggests that both the Central Spitsbergen Basin and the Arctic Ocean were largely cut off from the global oceans, with little evidence of an open Barents Shelf. The pre-PETM Svalbard strata does show an unradiogenic $^{187}Os/^{188}Os$ shift (Wieczorek et al., 2013), as also observed in this study (Table 2) and in North Atlantic datasets (Liu et al., 2019; Schmitz et al.,

2004). This consistent regional pattern suggests that a source of unradiogenic Os such as NAIP volcanism was also able to affect the northern part of the Norwegian-Greenland Seaway (Figure 1). The early Eocene shift to unradiogenic $^{187}Os/^{188}Os$ values at Lomonosov Ridge (Figure 5; (Dickson et al., 2015) may then indicate a post-PETM increase in seawater transfer between the Arctic and the peri-Tethys through western Siberia (Radionova et al., 2003).

The extreme change in $Os_i$ values from 0.3 to 0.83 at the beginning of the PETM recovery in the Danish strata (Table 2; Figure 5) is a large deviation from the global trend that continues for the entire Fur Formation. This radiogenic $Os_i$ shift occurs in samples 109 cm apart, or 88 cm excluding the thicknesses of Ashes -34 to -31 (Figure 3). Using the calculated sedimentation rates for Intervals 6 and 7 (*Sections 5.1.4 and 5.1.5*) gives an estimated duration of ~11.9 kyr

between the two samples, which suggests that a rapid and sustained isolation of the North Sea basin coincided with the start of the PETM recovery. This result corroborates previous findings of a freshening of the North Sea in the early Eocene (Zacke et al., 2009), while markedly improving the resolution of the timing of isolation to ~55.82 Ma. This timing is in excellent agreement with the estimated age of 55.80 ±0.8 Ma for a buried landscape surface in the Faroe-

Shetland basin (Hartley et al., 2011; Shaw Champion et al., 2008), which represented a major regional regression associated with NAIP uplift (Conway-Jones and White, 2022). The presence



of Ash -33 within the interval of rapid $^{187}Os/^{188}Os$ change is also noteworthy, as the source volcano is postulated to be the island of Lundy in southwest England (Larsen et al., 2003; Lisica et al., 2022). It is therefore plausible that the uplift associated with the NAIP reactivated the

Lundy volcanic system in conjunction with closing the English Channel, and/or that thick volcaniclastic deposits from the Lundy caldera clogged the proximal shallow seaway (Figure 1).

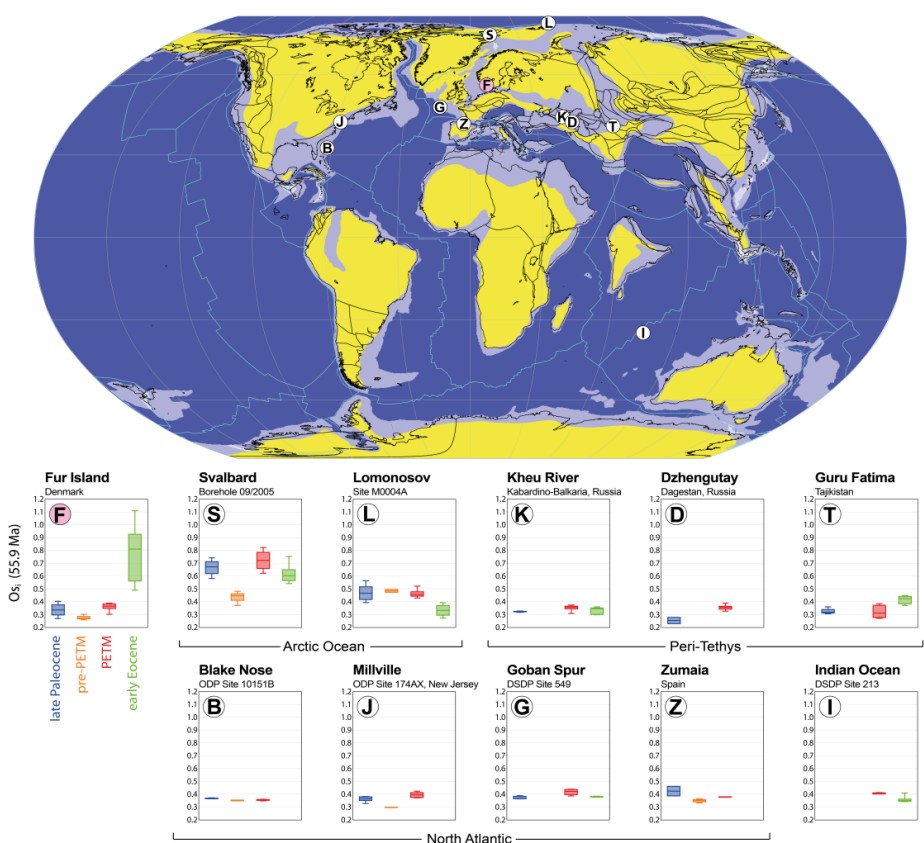

**Figure 7.** A global compilation of initial Os isotopes at 55.9 Ma (Os$_i$) from sites across the Paleocene−Eocene boundary. Fur Island (F, this study); Svalbard (S) (Wieczorek et al., 2013); Lomonosov Ridge (L), Kheu River in Russia (K), Dzhengutay in Russia (D), Guru Fatima in Tajikistan (T) (Dickson et al., 2015); Blake Nose (B), and

Millville in New Jersey (J) (Liu et al., 2019); Goban Spur site 549 (G), Indian Ocean Site 213 (I) (Ravizza et al., 2001); Zumaia in Spain (Z)* (Schmitz et al., 2004). *Os$_i$ calculated to 55 Ma in this study. Blue denotes late Paleocene samples, orange are confirmed pre-PETM data (last few kyr of Paleocene), red are during the PETM CIE, and green are post-PETM early Eocene samples. Plate tectonic reconstruction map adapted from Pogge von Strandmann et al. (2021).



### 5.2.2. Lipid biomarkers


Unlike all other analysed sites, the Lomonosov ACEX core and part of the Fur datasets do not show a marked response to the PETM-warming in fcren' (Figure 8). These observations suggest that Thaumarchaeotal membrane adaptation predominantly occurred through GDGTs 1–3 for these datasets and not also through increased fcren' as seen at other open marine localities. It


seems that only part of this divergent response can be ascribed to lower temperatures at both Fur and in the Arctic Ocean at the time, since $TEX_{86}$-derived SSTs as well as vegetation reconstructions favour warm-temperate to subtropical conditions at both localities (Sluijs et al., 2020; Suan et al., 2017; Weijers et al., 2007; West et al., 2015; Willard et al., 2019; Willumsen, 2004), which is similar to sites in the high-latitude Southern Hemisphere (Bijl et al., 2021;


Contreras et al., 2014; Huurdeman et al., 2020; Sluijs et al., 2011).

In addition, the relatively cold West Siberian Sea at ~58 ºN (Frieling et al., 2014) is considered to have had an open connection to the Arctic but also to the Peri-Tethys to the south, and its GDGT response follows the pattern of other globally distributed sites throughout the Paleocene and Eocene. Deoxygenation and fresh water input can be excluded as dominant controlling


factors since these are not unique to either the Danish or the Lomonosov strata (e.g. (Carmichael et al., 2017; Frieling et al., 2014; Sluijs et al., 2014). The divergent response between $TEX_{86}$ and fcren' to temperature changes in the latest Paleocene and early Eocene in the Fur and ACEX sites may therefore have been facilitated by temporary basinal restriction. It is intriguing that the $TEX_{86}$ – fcren' data from the Lomonosov locality shows a more normal marine response during


the Eocene Thermal Maximum 2 (~54 Ma) interval (Sluijs et al., 2020; Sluijs et al., 2009). While we cannot confirm a similar signal occurred in the North Sea area at the same time as this interval post-dates the Fur Formation, it highlights that exploring the behaviour of lipid-based proxies may aid in identifying and constraining periods of basin restriction in the region.

**Figure 8.** A comparison of $TEX_{86}$ and fcren' data across the PETM from existing global datasets: Fur Island (F)


(Stokke et al., 2020a), Lomonosov Ridge (L) (Sluijs et al., 2020), Turgay Straits in Siberia (Y) (Frieling et al., 2014), the Gulf of Mexico (M) (Smith et al., 2020), Bass River and Wilson Lake, New Jersey shelf (J) (Sluijs et al., 2007), ODP 959D on the Côte d'Ivoire margin (C) (Frieling et al., 2019), Nigerian margin BH10 (N) (Frieling et al., 2017), and ODP 1172D on the East Tasman plateau (E) (Sluijs et al., 2011). Blue denotes late Paleocene samples, orange are confirmed pre-PETM data (last few kyr of Paleocene), red are during the PETM CIE, and green are post-


PETM early Eocene samples. Plate tectonic reconstruction map adapted from Pogge von Strandmann et al. (2021).







### 5.3 Weathering and hydrology

The Li isotope record for the Stolleklint Clay and Fur Formation shows large variations both during and after the PETM (Figure 3). A major excursion in $\delta^7$Li values of -4‰, which is coeval

with the PETM onset and matches other global siliciclastic and carbonate Li isotope records, is interpreted to be a direct weathering response to an enhanced hydrological cycle during global warming (Pogge von Strandmann et al., 2021). However, the post-PETM record continues to show considerable variability, with three negative $\delta^7$Li excursions broadly coinciding with the start of Intervals 7, 8, and 9, each followed by positive excursions (Figure 3). Notably, the

excursion at the start of Interval 9 is of a comparable magnitude (-4‰) to the excursion during the PETM onset, albeit a gradual change over ~15 m rather than a sharp response over a 12 cm interval (Figure 6).

Numerous factors could have influenced the detrital $\delta^7$Li signal to generate the observed fluctuations in the Fur Formation. Increased surface temperatures elevate both chemical

weathering and erosion, and may increase erosion more than chemical weathering, leading to a lower weathering intensity regime, as proposed to explain the negative $\delta^7$Li excursion during the PETM onset (Pogge von Strandmann et al., 2021). However, the lack of correlation between Li isotopes and palaeotemperatures in post-PETM sediments suggests that climate was not the main driving factor of later $\delta^7$Li excursions (Figure 6). Alternatively, uplift associated with the

emplacement of the NAIP could affect detrital $\delta^7$Li values (Dellinger et al., 2017; Dellinger et al., 2015), since the increase in topography would steepen slopes and increase erosion rates, which could also lower the weathering intensity. This effect could account for the negative $\delta^7$Li excursion observed at the start of the PETM recovery (Interval 7), which was coeval with the NAIP uplift that isolated the North Sea from the North Atlantic Ocean. However, this NAIP

uplift phase was the last major regression that is recognised in proximal settings (Hartley et al., 2011; Shaw Champion et al., 2008; White and Lovell, 1997), so continued uplift events were unlikely to be responsible for the negative $\delta^7$Li excursions during Intervals 8 and 9 (Figure 6).

Variations in clay mineralogy and/or clay content could also cause $\delta^7$Li excursions in the bulk sediments, but there is little correlation between Li isotopes and clay mineralogy through the Fur

Formation (Figure 6). Peaks in kaolinite content, possibly indicating enhanced runoff transporting this denser clay further into the catchment (Nielsen et al., 2015; Stokke et al., 2021),



do not correlate with $\delta^7$Li values (Figure 6). The $\delta^7$Li excursions also do not appear to be linked to changes in the bulk lithology. The heavily laminated sections in Intervals 4 (Stolleklint Clay) and 8 (Fur Formation diatomite) show opposing $\delta^7$Li trends, while the upper sections of Interval

9 (unlaminated diatomite), where the clay component comprises 100% smectite, continue to show some isotopic variability (Figure 6). In summary, variations in clay mineralogy or lithology do not appear to be driving the observed $\delta^7$Li fluctuations.

One proxy that does broadly correlate with $\delta^7$Li variations in the post-PETM sediments is the $^{187}$Os/$^{188}$Os record (Figure 6). Osmium isotopes reflect a basin-wide signal, while the detrital

$\delta^7$Li record in Danish strata is likely representative of siliciclastic input from the Fennoscandian shield on the northeast North Sea margin (Anell et al., 2012). Assuming that the North Sea remained largely isolated from the North Atlantic Ocean during the deposition of the Fur Formation, the $Os_i$ record suggests an increase in basalt-derived fluxes towards the start of Interval 9 (Figure 6). The correlation with the decreasing $\delta^7$Li values suggests that volcanic ash

was likely being deposited on the Fennoscandian platform in increasing quantities towards the end of Interval 8, and that these ash deposits were rapidly weathered and eroded, forming isotopically light clays that were transported into the basin. During the 2010 eruption of Eyjafjallajökull in Iceland, clay formation occurred within just a few months of ash deposition in local catchments (Olsson et al., 2014; Paque et al., 2016; Pogge von Strandmann et al., 2019),

and this process was likely to have been faster in the warmer Eocene climate. The gradual $\delta^7$Li excursion of -4‰ from Interval 8 to Interval 9 also matches with an increase in Hg contents (Figure 3A) and estimated Hg accumulation rates (Figure 5), which is also consistent with a volcanic origin. In combination, these proxies suggest that the post-PETM variations in $\delta^7$Li values were largely governed by the rapid weathering and erosion of terrestrial ash deposits from

the NAIP, which reached a post-PETM peak around the start of Interval 9.

### 6. Conclusions

The combined proxies of volcanic ash deposition, clay mineralogy, Hg anomalies, Li isotopes, and initial $^{187}$Os/$^{188}$Os ($Os_i$) values indicate that NAIP activity is prevalent throughout the entire Paleogene strata exposed at Fur Island in northwest Denmark. However, the proxies wax and

wane, indicating significant changes in the NAIP activity across this interval (ca. 56–55 Ma). The late Paleocene Holmehus/Østerrende clay (Interval 1) contains a volcanogenic-rich



mineralogy and an unradiogenic $Os_i$ signature that suggest substantial erosion and weathering of basaltic terrains, while eruptive proxies suggest low-level but increasing NAIP activity towards the latest Paleocene. The strata above the possible unconformity (Interval 2) show more

unradiogenic $^{187}Os/^{188}Os$ values (mean $Os_i$ = 0.275), as observed in other pre-PETM records. The appearance of thick ash layers and Hg anomalies indicate a significant increase in NAIP activity just before the PETM. However, it is unclear what proportion of this activity was effusive, explosive, or volatile release from contact metamorphism around intrusions.

The PETM onset at Fur (Interval 3) is bounded by ash layers, but within the condensed $\delta^{13}C$

excursion there is little evidence for elevated NAIP activity. The PETM body (Intervals 4–6) has the highest sediment accumulation rates in the studied strata, and several proxies (e.g. $\delta^{13}C$; $^{187}Os/^{188}Os$) remain stable over ~24 m of stratigraphy. Mercury contents covary with organic carbon, resulting in near-constant Hg/TOC ratios. However, factoring in the elevated sedimentation rates, these data imply voluminous and continuous Hg accumulation rates ($Hg_a$)

that was more than an order magnitude greater than during both the PETM onset and recovery in the eastern North Sea. Combined with the sustained unradiogenic $Os_i$ values and a large increase in zeolite content, these findings suggest that the peak in NAIP activity occurred during the body of the PETM $\delta^{13}C$ excursion (ca. 55.9–55.8 Ma). The relative scarcity of ash layers in the PETM strata implies that this elevated activity was likely dominated by effusive eruptions and/or

thermogenic degassing.

The post-PETM Fur Formation (Intervals 7–9) contains the vast majority of the regionally recognized ash horizons, but other proxy evidence (e.g., $Hg_a$ rates) suggests an overall diminished NAIP activity. These findings suggest a change in eruptive style to more explosive activity, and therefore ash production, that was likely aided by increased magma-water

interactions within the rift system. Changes in detrital Li isotopes suggest that increased ash production enhanced silicate weathering and erosion fluxes, potentially increasing carbon sequestration during the PETM recovery (Longman et al., 2021) and early Eocene. These combined proxies indicate that much of the main acme of the NAIP activity, constrained to approximately 56–54 Ma based on the existing suite of sparse radiometric ages (Wilkinson et al.,

2017), was likely to have been concentrated in a much shorter interval between 56.0 and 55.8 Ma coincident with the PETM CIE.



There is evidence for transiently reduced salinity conditions in the North Sea and the Arctic Ocean in the late Paleocene and early Eocene, but the timing of these potential basin restrictions is not well constrained. A combination of Os isotopes and biomarkers were compared with

global datasets to provide high-resolution proxies for the restriction of the North Sea basin. The Os isotope record at Fur Island deviates rapidly from the global ocean signal in under 12 kyr at the end of the PETM body, coinciding with the start of the CIE recovery, which suggests that there was a rapid isolation of the North Sea contemporaneous with the end of hyperthermal conditions. The $TEX_{86}$ and fcren' records may indicate short-lived basinal restriction and/or

reduced salinity in pre-PETM strata, but the deviation from open-ocean datasets is most pronounced during the PETM recovery phase and early Eocene. Combined, these data suggest that the NAIP uplift closed the English Channel at an estimated 55.82 Ma, with the North Sea subsequently remaining restricted from the Atlantic Ocean for at least several 100 kyr. The close temporal correlation between the large radiogenic shift in Os isotopes and the end of the PETM

warrants further investigation, as it suggests a possible relationship between the NAIP uplift, seaway connectivity, and the end of hyperthermal conditions.

**Data availability**

All data generated in this study is available in the Supplementary data.

**Sample availability**

Samples may be available upon request.

**Author contributions**

MTJ, JF, HHS, and SP devised the study. MTJ, EWS, HHS, SP, TA, NT, MLV, CT, VZ, and BPS contributed to field work. MTJ, ADR, EWS, JF, PPvS, DJW, TA, NT, and TAM conducted laboratory analyses. MTJ prepared the manuscript with contributions from all co-authors.

**Competing interests**

The authors declare that they have no conflict of interest.

**Acknowledgments**

Gauti Eliassen, Lars Eivind Augland, Sara Callegaro, Olivia Jones, Christine Grabatin, and Claus Heilman-Clausen are warmly thanked for their assistance. This work was supported by the



Research Council of Norway through its Centres of Excellence funding scheme, project number
223272. MTJ and EWS were funded by the Research Council of Norway Ungeforskertalenter
project "Ashlantic", project number 263000. TAM and JF acknowledge funding from European
Research Council Consolidator Grant (ERC-2018-COG-818717-V-ECHO). PPvS was supported
by ERC grant 682760. DJW was supported by a NERC independent research fellowship
(NE/T011440/1). MLV was funded by the European Commission, Horizon 2020 project
ICECAP, grant no. 101024218.

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
