# Peer review of "Tracing North Atlantic volcanism and seaway connectivity across the Paleocene–Eocene Thermal Maximum (PETM)"

_EGUsphere, 2023_

## Community Comment (CC1)

Review for ***Tracing North Atlantic volcanism and seaway connectivity across the Paleocene-Eocene Thermal Maximum (PETM)*** by *Jones et al.*

General comment:

I really enjoyed reading this manuscript, it's thorough and very well-written. I believe it provides key insights into the timing and nature of NAIP activity during the PETM which may allow us to move forward in our understanding of the forcing behind this event. The proxies used complement each other well and may, hopefully, also serve as a basis for future studies into other LIP-related events. I appreciate the clear outlining of limitations and uncertainties, which make their final conclusions more robust. At this stage, my main 'issue' is that it can be a difficult paper to understand for non-specialists, due to the nature of the used proxies. I do believe it's an important paper for a wide audience and I think some small adjustments could already make it more accessible.

Specific comments:

*Lithium:* I am not an expert on the Li system, but I know that basalt weathering also contributes to the Li pool. Would it be possible to use Li in any way as an additional indicator for volcanism (together with (Os). Does your interpretation of $\delta^7Li$ in the paper take this into account or would it change if part of the signal is affected by basalt availability/emplacement?

*Seaway connectivity:* Does the timing/progression of North Sea restriction such as you find it show any links to changes in North Sea oxygenation (from Schoon et al., 2015)? Overall, it seems to me oxygen concentrations recover (or atleast deox. becomes less severe) in the Fur Formation. Could you add a few words on this, perhaps in section 5.2 where you list previously published evidence for the restriction?

*PETM C sources/sinks:* I personally would like to read more specifically if these conclusions can also inform us a bit about when different types of emissions ($CO_2$/$CH_4$) may have occurred and how the balance between source/sink shifted in this time interval (e.g. evidence for volcanism also during recovery and after PETM, coincident with climate recovery). I understand this might be a bit beyond the scope of the paper, but as the introduction goes into quite a bit of detail regarding sources/sinks, a few sentences on this would definitely be of interest.

*Line 752:* Would your conclusions be affected by the longer estimate (~170kyr) of Zeebe and Lourens (2019)? Is there a specific reason for choosing the shorter duration?

Technical corrections:

1. It would be helpful if the order in which proxies are described/shown is kept the same throughout the paper (e.g. first Hg, then osmium, etc.).
2. A short description of the clays you discuss, and their interpretation, is missing from section 3.4 Clay mineralogy
3. *Lines 65-68:* reference?

4. *Lines 83-85:* This sentence gives the impression that constraining volumes/fluxes is the goal of the paper, so it may need to be reformulated to avoid this.
5. *Figure 1:* the color for subaerial volcanism looks more pink in the figure, while in the legend it appears more orange. Check if it's indeed the same.
6. *Figure 1:* A bigger contrast in colors between subaerial volcanism and volcanic centers would also be welcome.
7. *Figure 3:* What does the red space in the d13C panel indicate? It's also not listed in the caption.
8. *Figure 3:* As I mentioned above, the number and complexity of the proxies that have been used (esp. Hg, Li and Os) make this paper somewhat difficult to read for non-experts. I was wondering if mechanism indicators could be added to this figure? Something like an arrow with more volcanism/more weathering on either side. Otherwise, a table with the proxies and their interpretations could also work and give space for a bit more nuance than such an arrow might. If there's space within the paper, I think it would help a lot.
9. *Lines 702-703:* What does 'waning' mean in this case? Less overall explosive volcanism (and shift to more effusive) or a shift from more explosive to less explosive volcanism (e.g. because of less hydromagmatic interactions?)? Could this be clarified?

---

## Community Comment (CC2)

**Comment on: "Tracing North Atlantic volcanism and seaway connectivity across the Paleocene–Eocene Thermal Maximum (PETM)" in EGUsphere**

Following the recent public comments made by Jones et al. (in review) regarding our recent paper (Gernon et al., *Nat. Geosci.*, 2022; https://doi.org/10.1038/s41561-022-00967-6), we feel compelled to respond, as we have identified some key errors and misrepresentations of our work in their comments. Unfortunately, the authors did not communicate their specific concerns directly to us prior to posting or alert us to their public comments on our study. Our primary concern here is to ensure the accurate representation of our work. In this Comment, we address the specific text we take issue with, presented by those authors in response to a review by Marcus Gutjahr, whilst also providing some important clarifications.

Firstly, Jones et al. state *"There are critical geochronological and geochemical issues with this study (briefly outlined below) that impact the conclusions and viability of the proposed concept of extreme sub-crustal carbon release coincident with the PETM. Discussing these issues is outside the scope of our present manuscript and hence we prefer not to cite this work."*

[1] Jones et al. then assert that *"the cited sedimentation rates of 50 cm kyr$^{-1}$ that form the basis for the PETM duration in the Rockall section do not match the site report's estimate of 9.5 cm kyr$^{-1}$ with a maximum of 26 cm kyr$^{-1}$ in the section. As such, the PETM interval is based on biostratigraphic observations of a single dinoflagellate cyst marker species in a single sample (site report), so the duration of this critical interval of the succession at Site 555 cannot currently be constrained with confidence."*

This is incorrect: while the absolute age models have of course been updated, we urge the authors to cross-check the depths in DSDP Leg 81 Hole 555 referred to in Extended Data Fig. 1 of our paper. Here, they will see that the inferred position of the PETM boundary between the 600 to 700-metre range falls firmly within strata exhibiting sediment accumulation rates of approximately 50 cm/1000 yr, as evidenced in Figure 2 of the site report by Backman et al. (1984), referred to in our paper.

[2] The authors further comment: *"The presented ages of the East Greenland and Faroe Islands basalts have not been corrected to the most recent 28.201 Ma Fish Canyon Tuff calibration (Kuiper et al., 2008). Recalculating these ages gives 56.78 ± 0.25 Ma for the base of the Milne Land Fm in East Greenland (from 56.1 ± 0.4 Ma), which places the chemical heterogeneities observed in the basalts much earlier than the PETM interval. The ages of the Faroe Islands lavas are heavily debated, with studies arguing that the PETM interval could be above or below the hiatus. A recalibrated Ar/Ar age of 55.57 ± 0.35 Ma from the Middle basalt series in the Faroe Islands (Storey et al., 2007b) gives a post-PETM age for these lavas. This result suggests that the post-hiatus, high-Ti basalts observed in East Greenland and the Faroes may not be synchronous. This implies, with the current best estimates, that the analysed materials span at least 1 Myr, not ~200 kyr as proposed."*

It is unclear to us, (1) how Jones et al. arrived at a revised age of 56.78 ± 0.25 Ma for the base of the Milne Land Formation; (2) what is reported in the age uncertainty (i.e., analytical uncertainty or decay constant uncertainty?), and (3) what is the confidence level of the reported uncertainty. When comparing age data from multiple chronometers (e.g., Ar/Ar and astronomical techniques) it is critical that age uncertainties include all sources of uncertainty and are reported at the 2-sigma confidence interval for data comparison.

When we recalculate the Ar/Ar age of 56.1 ± 0.4 Ma (2 sigma, analytical uncertainty) (Storey et al., 2007) using the Kuiper et al. (2008) Fish Canyon Tuff sanidine calibration (28.201 Ma), which uses the decay constants of Min et al. (2000), we obtain an updated Ar/Ar age of 56.466

± 0.403/0.434 Ma (2 sigma, internal/external precision), which is 0.31 ± 0.47 Ma younger relative to the age presented by Jones et al. Importantly, due to the large analytical uncertainties associated with the Storey et al. (2007) measurements, this recalculated age is indistinguishable (0.36 ± 0.59 Ma) from the age determined by Storey et al. (2007), and considering all sources of uncertainty, is 0.45 ± 0.43 Ma older than the onset of the PETM (using the 56.01 ± 0.05 Ma astronomical age determination of the PETM from Zeebe and Lourens, 2019). Contrary to Jones et al., these revised ages allow the stratigraphically lower volcanism of the Middle Lava Formation to be closely related with the PETM onset, which is consistent with the model proposed by Gernon et al. (2022).

There are similar issues with regards to the updated age determination (55.57 ± 0.35 Ma) for the Middle lava series as reported by Jones et al, and as they have not provided the original age used as reported by Storey et al. (2007b), we cannot verify their result. On face value this age presented by Jones et al. is 0.44 ± 0.35 Ma younger than the PETM, which is again not inconsistent with Gernon et al.

The use of the astronomical tuned calibration of the Ar/Ar system (Kuiper et al. (2008) is most appropriate here as we are comparing Ar/Ar ages with an astronomical age for the PETM, but we need to consider the limitations of this calibration. The Kuiper et al. calibration only addresses inaccuracies in the age of the Ar/Ar standard and not inaccuracies in the decay constant of the Ar/Ar system and there remains an ongoing issue with regards to inter-chronometer comparisons due to the difficulties in assigning quantitative uncertainties to astronomical ages that underpin the calibration.

Taking into account these factors, the authors cannot discount the possibility that the lava sequence is synchronous with the PETM. Moreover, there is clearly a false dichotomy in their argument: we have already shown—using radioisotopic ages and critical regional geological evidence that may have been overlooked by the authors, e.g., magnetic polarity chrons—that many of the Middle Lavas in the Faeroes are likely post-PETM (refer to Fig. 2b-c in Gernon et al. (2022)) — it is a very thick sequence. Taken together, these lines of evidence do not preclude synchronicity with the PETM, nor (considering all sources of age uncertainty) do they support Jones et al.'s assertion that the relevant interval "spans 1 million years".

[3] Reviewer Gutjahr had merely suggested that the authors might consider the realistic possibility of a higher mantle-derived $CO_2$ release scenario (8% or more) from the subcontinental lithospheric mantle. Jones et al. respond that *"The authors choose a pre-eruptive $CO_2$ concentration of 2 wt% for flood basalt eruptions in their model, citing Self et al. (2005), despite this cited paper stating "...0.5 wt% [is] a reasonable but possibly high value for pre-eruptive $CO_2$ concentration [in flood basalt eruptions]". The Monte Carlo calculations assume concentrations ranging from 1–8 wt%, all in excess of this value. There is no convincing geochemical evidence from the northeast Atlantic margin that currently supports such elevated $CO_2$ concentrations."*

Unfortunately, the authors have made another key error here. The section on 'Quantifying background volcanic $CO_2$ fluxes' of Gernon et al. (2022) focuses on determining the *maximum* potential $CO_2$ release from ridges and Large Igneous Provinces (or LIPs) to assess their potential contribution to PETM warming. To estimate this value, we considered pre-eruptive $CO_2$ concentrations of 2 wt%, which is perfectly reasonable given our overarching motivation in this part of our study to assess the *maximum* likely outputs from LIPs under a 'business as usual' scenario.

We respectfully note that the main numerical model of Gernon et al. (2022) [refer to Fig. 3 in that paper] is fully described in the paper's Methods section, which may not have been fully

considered in the authors' critique. Here we state that we use "a Beta distribution with a mean value of 0.5 wt%, and minimum and maximum values of 0.2 wt% and 2 wt%", which as Jones et al. themselves acknowledge is a perfectly reasonable pre-eruptive $CO_2$ content for flood basalts (Self et al, 2005). Jones et al. are therefore incorrect in asserting that our "calculations assume concentrations ranging from 1–8 wt%, all in excess of this value [0.5 wt%]". Unfortunately, the authors appear to be conflating this with a separate model run, in which we evaluate the probable carbon release from the sub-continental lithospheric mantle (SCLM) keel. In this model, our chosen values of 1–8 wt% are completely consistent with, and in fact more conservative than, the expected range for the metasomatized SCLM (see for example, Foley and Fisher, *Nat. Geosci.* 10, 897–902, 2017).

While it is of course their prerogative not to cite Gernon et al. (2022), which presents an alternative point of view, their arguments against citing this work seem to be based on incorrect interpretations rather than objective facts or sound reasoning.

**Dr Tom Gernon** and **Prof Martin Palmer**
University of Southampton, National Oceanography Centre, Southampton, United Kingdom.
**Dr Dan Barfod** and **Prof Darren Mark**
Scottish Universities Environmental Research Centre, Scottish Enterprise Technology Park, Glasgow G75 0QF

---

## Author Response (AR1)

Morgan Thomas Jones
Email: m.t.jones@geo.uio.no
Postbox 1028 Blindern
N-0315 Oslo, Norway
Phone: +47 97148221

20th June 2023

Dear Yannick Donnadieu,

It is our pleasure to resubmit the following manuscript entitled "Tracing North Atlantic volcanism and seaway connectivity across the Paleocene–Eocene Thermal Maximum (PETM)" for consideration at Climate of the Past. The main text comprises 14,916 words, including the abstract and figure captions. In addition, there are eight figures and two tables, plus the raw data available as an appendix.

We have addressed all of the suggestions by the two reviewers, as detailed in the responses to their reviews on *EGUsphere*. Those responses are copied here for ease of access. The only changes that were not tracked in the track-changes file were the suggestions of alterations to the figures themselves. In addition, based on your feedback, we now include two sentences referencing the work of Gernon et al. (2021) in the introduction (lines 78–84), and we have edited the response to Marcus Gutjahr's comment below accordingly. We hope that these minor revisions are sufficient for publication in *Climate of the Past*. We look forward to hearing from you at your earliest convenience.

Yours sincerely,

Morgan Jones

**Marcus Gutjahr review:**

In this manuscript under discussion Morgan Jones and co-authors present a comprehensive set of geochemical and isotopic data in order to reconstruct regional and global aspects of NAIP activity, hydrological changes, weathering, and seawater connectivity across the PETM in an outstanding extended sedimentary succession on Fur Island in Denmark. I particularly like the combined multi-proxy study of proxies for volcanic activity (e.g. Hg/TOC, Hg anomalies, Os isotopes), temperature proxies, and chemical weathering indicators (Li and Os isotopes). The authors discuss all these geochemical sedimentary parameters on an outstanding sedimentary section in relative proximity to the North Atlantic Igneous Province. It is arguably a long manuscript, maybe here and there the discussion could be a little shorter, but overall this work is well prepared, very well written and appears quite polished. I have no major comments, but several moderate and minor, which I raise as presented in the manuscript.

*We thank the reviewer for their thorough and fair assessment of the manuscript. We have addressed all of their points below (in italic).*

Lines 78-81:

Here the authors state: "Yet, this high-volume carbon release scenario might be at odds with the extremely enhanced organic carbon burial rates for the PETM, a carbon sink would rapidly drive exogenic δ13C to positive values if not balanced by a heavily 12C-enriched source…"

I find this statement a little puzzling. In our mentioned study (Gutjahr et a., 2017, Nature) we clearly showed that – despite our modelled very high carbon emission rates over the CIE – enhanced organic carbon burial following the peak CIE is required for our intermediate complexity model to track the marine d13C evolution as recorded in post-CIE planktic foraminifera. In other words, without enhanced organic carbon burial, our geochemical data could not be brought into agreement with the cGENIE model output. Therefore our most realistic carbon release budget was on the order of 12,200 Gt C.  I agree, however, that the required total budget of additional organic carbon burial still is, and will be, a matter of debate for some time.

*It is beyond the scope of this paper to go into too much detail here, but the recent study from Papadomanolaki et al. (2022) highlighted the issue that most modelling studies require a reduction in organic carbon burial during the CIE body to sustain the extreme $\delta^{13}C$ conditions, and this is not observed in the geological record. The study of Gutjahr et al. (2017) is a seminal paper and advances our understanding of carbon cycle dynamics during the PETM, but some issues remain. For example, their modelled CIE body is shorter than estimated from sedimentary sections, and organic carbon burial starts 30 kyr after the CIE onset. Based on the reviewer's comments, we have rephrased this section to the following:*

*"However, most carbon cycle model scenarios appear to be at odds with the extremely enhanced organic carbon burial rates during much of the PETM (Kaya et al. 2022, John et al. 2008). A recent modelling study demonstrated that a large organic carbon sink would rapidly drive exogenic $\delta^{13}C$ to positive values unless the impact of organic carbon burial was reduced during the CIE body (Papadomanolaki et al., 2022). More often, a scenario is chosen that focuses the impact of organic carbon burial to the later parts of the CIE to match the CIE recovery (Bowen & Zachos, 2010; Bowen, 2013; Gutjahr et al., 2017; Papadomanolaki et al., 2022). However, scenarios with reduced organic carbon burial during the initial phases of the CIE are in conflict with field observations (e.g. John et al. 2008, Kaya et al. 2022), complicating interpretations of the CIE purely on grounds of the source $\delta^{13}C$ signature."*

90-92:

Gernon et al. (2022, Nature Geoscience) recently alternatively suggested release of mantle-derived carbon from the subcontinental lithospheric mantle with much higher $CO_2$ concentrations of 8% or more.

*On the recommendation of the editor, we have now included this reference, but we have highlighted that the current geochronology that the model is based on is flawed. The following text has been added to the introduction:*

*"A recent study by Gernon et al. (2021) suggested elevated magmatic carbon release from a lithospheric mantle source may have augmented NAIP degassing fluxes during the PETM. However, their model is based on localities with sparse, uncorrected, and ambiguous geochronological data (e.g. Passey and Jolley, 2008; Wilkinson et al., 2017), and a refined and up-to-date bio- and chemostratigraphic control of target localities is required for this hypothesis to be thoroughly tested."*

Figure 2B:

The d18O and d13C data shown here are not from this section, right? Could the authors please make this clearer in the figure? I initially thought these would be local stable isotope records.

*We have added the headers "Global isotopic records" and "Regional stratigraphy" to figure 2B to remove ambiguity.*

260:

Do the authors have any idea towards the origin of this glauconite-rich silty horizon at the base of the CIE? Presuming these are authigenic in origin, what conditions would have been needed to allow formation of this glauconite layer?

*The formation of the glauconite-rich horizon is mainly authigenic in origin, suggesting extended exposure at the sea floor and indicative of slow sediment accumulation rates or a hiatus in deposition. We have added the following sentence to the manuscript:*

*"The transition from the Holmehus/Østerrende Formation to the Stolleklint Clay is marked by a possible hiatus of unknown duration and a glauconite-rich silty horizon (Heilmann-Clausen, 1995; Schmitz et al., 2004), which is comprised of mainly authigenic grains and is interpreted as evidence of very low sedimentation rates (Schoon et al., 2015). Above this glauconite-rich horizon, there is no clear evidence of any breaks in sedimentation until the top of the Fur Formation (Heilmann-Clausen et al., 1985; 2014; Stokke et al., 2020a)."*

Section 3.3.:

What about total procedural blank levels for Li, and did the authors report any secondary Li isotope standard results?

*We have added the following text to section 3.3:*

*"The measured δ7Li values secondary standards at this facility are Seawater: 31.17 ± 0.38‰ (n=43); USGS BCR-2: 2.57 ± 0.30‰ (n=11); USGS SGR-1b: 3.82 ± 0.28‰ (n=9). The total procedural blank is ≤ 0.003 ng Li (Pogge von Strandmann et al., 2019)."*

461:

Is fcren' hence a qualitative proxy for warm and saline waters?

*It could potentially be used in such a way, but investigating the pros and cons of this is beyond the scope of this paper. This is why we use it as a supporting argument to the Os isotopes for investigating seaway connectivity rather than as a standalone feature.*

Figures 3, 5, 6, 7 and throughout the text:

Please do not use Os(i) as axis title for a 187Os/188Os isotopic composition. Better use 187Os/188Os(i) (all with super-/sub-scripts respectively). Same goes for its usage throughout the manuscript. Just using Os(i) creates the wrong impression that we are dealing with an elemental proxy.

*We have replaced $Os_i$ with $^{187}Os/^{188}Os_{(i)}$ in figures 3, 5, 6, and 7, and throughout the text.*

Section 4.2:

I find the evolution in Hg content (both elemental and relative to TOC) striking in that no peak is seen at the interval with most abundant ash layers. This could suggest that the type of volcanism (sub-marine vs sub-aerial) may have quite some impact on Hg abundance. And the general pattern of Hg abundance makes it appear like a very general proxy for volcanic activity, but I may be wrong. The authors discuss the Hg evolution in section 5.1., which is appreciated. I'd also be interested to know whether the almost anti-correlated ash layer abundance vs. Hg concentration peaks simply track these different styles of volcanism that were encountered during the emplacement of the NAIP. Could the authors expand a bit more on this feature? Is there a good understanding in the PETM NAIP literature as to the importance of sub-marine as opposed to sub-aerial volcanism, or transitions from one phase to another? This certainly ought to have an impact on geochemical records such as presented here. This is already slightly touched upon in section 5.1 but could be expanded.

*We agree with the reviewer that the Hg proxy is complicated, and that it is striking that there is not a significant enrichment associated with the positive ash series. The increase in Hg anomalies with increasing proximity to the NAIP was covered in detail by Jones et al. (2019), but we have added to section 5.1in this manuscript to draw the reader's attention to how this complex proxy was affected by changing volcanism/magmatism across the PETM. The subaerial vs submarine emissions appears particularly important for the atmospheric dispersal of Hg. There is compelling evidence from modern systems that submarine degassing of Hg significantly reduces atmospheric fluxes, and restricts Hg deposition in enclosed marine environments to just 10s km from the source in passive degassing scenarios (Tomiyasi et al., 2007). This suggests that submarine emissions are going to be heavily restricted in terms of their geographical distribution, and that the explosivity of the eruption or hydrothermal vent will have a huge impact on subsequent Hg dispersal. We have added the following text to section 5.1:*

*"The regional distribution of Hg emissions would be heavily affected by whether Hg degassing is subaerial or submarine (Jones et al., 2019a; Percival et al., 2018). Passive submarine degassing around modern volcanic systems can limit Hg deposition to just 10's km from the source in enclosed, shallow marine environments (e.g. Tomiyasi et al., 2007), so the depth and explosivity of submarine emissions will have a huge impact on subsequent Hg dispersal."*

621:

"Phreatomagmatic" is probably a more accurate term than "hydro-magmatic".

*We have changed it to phreatomagmatic.*

674-677:

Do unradiogenic Os isotope ratios really only reflect elevated basalt weathering, or could this also be partially controlled by direct Os release to seawater during sub-marine volcanism or hydrothermal activity? The PETM core section with these unradiogenic Os isotope compositions looks remarkably constant compared with the sections above and below.

*We have changed the text to include all mantle-derived sources:*

*" These values imply that mantle sources were already a major component of global Os fluxes, including basalt weathering from ongoing NAIP activity and the earlier tropical emplacement of the Deccan Traps at 66.5–65 Ma (Schoene et al., 2019)."*

727-731:

With regard to the compositional change in Os isotopes during the PETM section, the authors mention the potential importance of thermogenic release of Os during contact metamorphism. Above in the same paragraph the authors correctly discuss the importance of weathering feedbacks controlling the Os isotopic shift. This part here with the thermogenic contribution seems a bit desperate. It may play a role, yet is not really required, is it?

*The reviewer is correct that it is not required to explain the observed Os isotopic shift, so we have deleted the second part of the sentence and removed the reference to Dubin (2015). The sentence now reads:*

*"The shift to radiogenic $^{187}Os/^{188}Os$ values suggests that an increase in submarine volcanism is unlikely during the onset."*

Section 5.2.1

Good discussion. But before comparing seawater Os isotope data from different ocean basins with regard to their comparability I consider it mandatory to introduce to what extent these literature sourced compositions indeed reflect an original past seawater signature that is not altered by detrital contributions in the sediment. Recovering a past seawater Os isotope signal from marine sediments may not be trivial at all, therefore such a preliminary assessment must be done. If sediments were completely digested to extract its Os isotopic signal these always contain a detrital component.

*We thank the reviewer for pointing out this omission in our discussion. We have added an introductory paragraph at the beginning of Section 5.2.1 addressing this important point:*

*"There are now several $^{187}Os/^{188}Os$ data sets from numerous global localities that can be used to assess the extent of ocean connectivity during the latest Paleocene and early Eocene (Figure 7). The methodology of Os retrieval has evolved through time. Older PETM studies used inverse aqua regia for sample digestion (e.g. Dickson et al., 2015; Ravizza et al., 2001; Schmitz et al., 2004; Wieczorek et al., 2013), while more recent analyses used chromic acid to preferentially liberate hydrogenous Os*

*(Liu et al., 2019; this study). The inverse aqua regia digestion method is more aggressive, potentially leading to contamination from detrital Os in silicate minerals (e.g. Kendall et al., 2004). However, the existing $^{187}Os/^{188}Os$ data across the PETM is largely homogeneous across the North Atlantic, Indian, and Tethys Oceans (Dickson et al., 2015; Liu et al., 2019; Ravizza et al., 2001; Schmitz et al., 2004). This global homogeneity suggests that any detrital contamination is minimal, and that the open marine Os residence time exceeded the ocean mixing time."*

Section 5.3

The Li isotopic results were probably a little sobering for the authors. When I first saw the detrital d7Li record in Pogge von Strandmann et al. (2021, Science Advances) I was quite excited to see such apparent dramatic climatic Li isotopic signatures being preserved in shallow marine sediments over the PETM CIE. Now being able to see the bouncing of Li isotopic compositions over the entire section makes the initial changes look less significant and (playing Devil's advocate) in the worst case coincidental. The discussion here is fine, but I have the impression we still need to understand the underlying controls in generating lithogenic Li isotopes better.

*On the contrary, we found the lithium isotopic results exciting! The study of Pogge von Strandmann et al. (2021) highlighted the perturbation of ocean Li isotopes using three carbonate localities, as well as large Li isotope excursions from two siliciclastic, shallow marine localities. These epicontinental sea sections are affected by numerous processes, so the clear signal of increased weathering at the start of the PETM is a key finding. The continued perturbation to the Li isotopic system into the Eocene is not that surprising, given the changes to regional uplift, subsidence, eustatic sea level change, flood basalt volcanism, hydrothermal venting, and explosive eruptions. An important consideration is the duration of these disturbances. The -4‰ $\delta^7Li$ excursion at the PETM onset occurred <10 kyr, while the -4‰ $\delta^7Li$ excursion encompassing the start of the positive ash series took place over an estimated 160 kyr. This suggests a more gradual response to changing weathering conditions into the eastern North Sea catchment, such as an increase in ash-forming, explosive eruptions that increased silicate weathering through enhanced particle surface area. We have added the estimated durations of the $\delta^7Li$ excursions to the first paragraph of section 5.3.*

Review for Tracing North Atlantic volcanism and seaway connectivity across the Paleocene Eocene Thermal Maximum (PETM) by Jones et al.

General comment:

I really enjoyed reading this manuscript, it's thorough and very well-written. I believe it provides key insights into the timing and nature of NAIP activity during the PETM which may allow us to move forward in our understanding of the forcing behind this event. The proxies used complement each other well and may, hopefully, also serve as a basis for future studies into other LIP-related events. I appreciate the clear outlining of limitations and uncertainties, which make their final conclusions more robust. At this stage, my main 'issue' is that it can be a difficult paper to understand for non-specialists, due to the nature of the used proxies. I do believe it's an important paper for a wide audience and I think some small adjustments could already make it more accessible.

*We thank the reviewer for their positive feedback and thorough review. We have addressed all of their points below (in italic).*

Specific comments:

Lithium: I am not an expert on the Li system, but I know that basalt weathering also contributes to the Li pool. Would it be possible to use Li in any way as an additional indicator for volcanism (together with Os). Does your interpretation of $\delta^7$Li in the paper take this into account or would it change if part of the signal is affected by basalt availability/emplacement?

*As it is a stable isotope system, the lithium isotopic signature in the sediments is primarily affected by clay formation and dissolution. Therefore, the emplacement of easily-weathered fresh volcanic material is likely to drive siliciclastic $\delta^7$Li values to lower values, which indeed seems to be the case with sections of this dataset.*

*The large negative $\delta^7$Li excursion at the PETM onset is likely to be a response to extreme warming, which would have increased chemical weathering through an enhanced hydrological cycle (see Pogge von Strandmann et al., 2021). However, the slight radiogenic shift in Os isotopes during the*

*earliest phases of the PETM suggests that increased weathering of volcanic material was not the cause of the initial $\delta^7Li$ excursion.*

*In contrast, the post-PETM $\delta^7Li$ variations appear to covary with Os isotopes, with low $\delta^7Li$ values coinciding with unradiogenic Os isotopes, which supports the weathering of fresh volcanic material as a primary driver of the observed changes in these proxies. Both of these isotope systems are likely to be sensitive to changes between effusive and explosive activity (even if the latter is volumetrically less significant) because the formation of ash and scoria leads to an increase in the particle surface area by several orders of magnitude compared to lavas. Therefore, the large but gradual shift in both Li and Os isotopes towards the positive ash series may reflect a shift from effusive- to explosive-dominated volcanism.*

Seaway connectivity: Does the timing/progression of North Sea restriction such as you find it show any links to changes in North Sea oxygenation (from Schoon et al., 2015)? Overall, it seems to me oxygen concentrations recover (or at least deox. becomes less severe) in the Fur Formation. Could you add a few words on this, perhaps in section 5.2 where you list previously published evidence for the restriction?

*It is certainly plausible that the restriction of the North Sea had an effect on marine anoxia. However, the Schoon et al (2015) redox record does not continue into the post-PETM strata. Stokke et al. (2021) show that S, Mo, and U contents all decrease during the PETM recovery, suggesting a reduction in anoxic conditions. However, it is difficult to separate the regional effects of the North Sea isolation from the global oceans, and the recovery from marine anoxic conditions induced by the PETM.*

PETM C sources/sinks: I personally would like to read more specifically if these conclusions can also inform us a bit about when different types of emissions ($CO_2$/$CH_4$) may have occurred and how the balance between source/sink shifted in this time interval (e.g. evidence for volcanism also during recovery and after PETM, coincident with climate recovery). I understand this might be a bit beyond the scope of the paper, but as the introduction goes into quite a bit of detail regarding sources/sinks, a few sentences on this would definitely be of interest.

*There are a few papers in preparation that will deal with this issue more directly, based on material from the recent IODP Expedition 396 on the Norwegian continental margin. In short, it is not straightforward to draw any clear conclusions regarding changes in C balance based on the currently*

*available data, so we would prefer not to go into too much detail here. A couple of sentences can be added to the conclusions to briefly touch on this subject if required.*

Line 752: Would your conclusions be affected by the longer estimate (~170kyr) of Zeebe and

Lourens (2019)? Is there a specific reason for choosing the shorter duration?

*Although a longer PETM body duration will decrease our estimates of mass accumulation rates by ~40%, the sedimentation rates compared to pre-PETM and post-PETM strata are still an order of magnitude higher, which implies that our conclusions would not be affected.*

*We chose a PETM onset age of 55.93 Ma (Westerhold et al., 2017) as this cyclostratigraphic age appears to be the better fit with existing geochronological data. In particular, the precise U-Pb age of 55.785 ± 0.034 Ma from a bentonite within the PETM carbon isotope excursion (CIE) from the Longyearbyen section in Svalbard (Charles et al., 2011) appears to be incompatible with an onset age of 56.01 ± 0.05 Ma **and** a PETM body duration of 170 ± 30 kyr (Zeebe and Lourens, 2019). It is a little difficult to discern where in the CIE the bentonite is in the Svalbard strata, as there is no obvious inflection between the body and recovery phases. However, the ash layer is still within the zone of elevated concentrations of Apectodinium Augustum (Charles et al., 2011), which suggests that it is indeed part of the CIE body or early in the recovery. Therefore, the 145 kyr difference between this bentonite age and the cyclostratigraphic age of Westerhold et al. (2017) is a better fit than the 225 kyr difference between the Zeebe & Lourens (2019) onset age and this syn-PETM ash layer.*

Technical corrections:

1. It would be helpful if the order in which proxies are described/shown is kept the same

throughout the paper (e.g. first Hg, then osmium, etc.).

*We carefully revisited the manuscript structure to see if there were ways to improve clarity for the reader. The proxies are described in the same order in the Methods, Results, and Figures, while in the Discussion these proxies are applied to specific time intervals and events (such as seaway connectivity).*

2. A short description of the clays you discuss, and their interpretation, is missing from

section 3.4 Clay mineralogy

*We did not go into detail describing the clays because this was presented in Stokke et al. (2021), but we can add a brief synthesis of those findings if required.*

3. Lines 65-68: reference?

*We have added a reference to Storey et al. (2007a).*

4. Lines 83-85: This sentence gives the impression that constraining volumes/fluxes is the goal of the paper, so it may need to be reformulated to avoid this.

*We have reformulated this sentence to:*

*"Constraining the timing and style of NAIP activity is critical to understanding the volumes and fluxes of each potential carbon source, in order to determine their roles in the initiation and long duration of the PETM."*

5. Figure 1: the colour for subaerial volcanism looks more pink in the figure, while in the legend it appears more orange. Check if it's indeed the same.

*There is indeed a mismatch between the key and the figure. We have edited the figure to standardise the colours.*

6. Figure 1: A bigger contrast in colours between subaerial volcanism and volcanic centres would also be welcome.

*A good suggestion, this was combined with the suggestion above to increase the contrast.*

7. Figure 3: What does the red space in the d13C panel indicate? It's also not listed in the caption.

*It shows the PETM carbon isotope excursion, as labelled by the column just to the left. We have added the following text to the Figure caption to improve clarity.*

*"The $\delta^{13}C_{org}$ data are from previous studies (Jones et al., 2019a; Schoon et al., 2013), with the red infill denoting the PETM CIE."*

8. Figure 3: As I mentioned above, the number and complexity of the proxies that have been used (esp. Hg, Li and Os) make this paper somewhat difficult to read for non-experts. I was wondering if mechanism indicators could be added to this figure? Something like an arrow with more volcanism/more weathering on either side. Otherwise, a table with the proxies and their interpretations could also work and give space for a bit more nuance than such an arrow might. If there's space within the paper, I think it would help a lot.

*Arrows have been added to the Li and Os isotope systems to improve clarity for non-experts in Figure 3, as Li isotope changes are dominated by clay formation whereas Os isotopes are dominated by mixing between unradiogenic (mantle) and radiogenic (continental) end members. On the other hand, Hg is quite a complex system without distinct end-member behaviour, and isolated Hg anomalies do not always indicate an increase in Hg input (such as increased volcanism). We cover this interpretation in detail in the text, so we believe that adding arrows for Li and Os isotopes, but not Hg, is the most appropriate way forward.*

9. Lines 702-703: What does 'waning' mean in this case? Less overall explosive volcanism (and shift to more effusive) or a shift from more explosive to less explosive volcanism (e.g. because of less hydromagmatic interactions?)? Could this be clarified?

*We intended to convey that there is very little evidence of explosive volcanism during the PETM body at Fur, which means that either the magnitude of explosive eruptions decreased (thereby not transporting ash as far as Denmark) and/or there was a decrease in explosive activity (so that fewer ash producing events occurred). We have improved the text to avoid confusion:*

*"The PETM onset at Fur is bounded by ash SK2 and the much thinner SK3 and SK4 layers that are the last ash occurrences for >19 m of strata (Figure 3), suggesting that explosive volcanism either decreased in magnitude to not transport ash as far, or that there was a period of explosive volcanic quiescence."*

*An important point to consider is that the presence/absence of ash layers does not inform on whether this could be a switch to more effusive activity. The NAIP is large enough that effusive and explosive activity could be occurring in different parts of the province at the same time. That said, the high Hg deposition rates at Fur during the PETM, coupled with the field evidence of >5 km of flood basalts erupted in East Greenland and radiometrically dated to within the PETM interval, provides strong evidence that intense effusive activity did indeed occur during the PETM.*